



# New Particle Formation dynamics in the central Andes: Contrasting urban and mountain-top environments

Diego Aliaga[1], Victoria A. Sinclair[1], Radovan Krejci[2], Marcos Andrade[3,4], Paulo Artaxo[5], Luis Blacutt[3], Runlong Cai[1,6], Samara Carbone[7], Yvette Gramlich[8], Liine Heikkinen[1,2], Dominic Heslin-Rees[2], Wei Huang[1], Veli-Matti Kerminen[1], Alkuin Maximilian Koenig[3,9], Markku Kulmala[1,10,11], Paolo Laj[1,12], Valeria Mardoñez-Balderrama[3,13], Claudia Mohr[2,8], Isabel Moreno[3], Pauli Paasonen[1], Wiebke Scholz[14], Karine Sellegri[15], Laura Ticona[3], Gaëlle Uzu[15], Fernando Velarde[3], Alfred Wiedensohler[16], Doug
Worsnop[1,17], Cheng Wu[2,a], Chen Xuemeng[18,19], Qiaozhi Zha[10], Federico Bianchi[1]

[1]Institute for Atmospheric and Earth System Research, Physics, Faculty of Science, University of Helsinki, Helsinki, Finland
[2]Department of Environmental Science & Bolin Centre for Climate Research, Stockholm University, Stockholm, Sweden
[3]Laboratory for Atmospheric Physics, Institute for Physics Research, Universidad Mayor de San Andrés, La Paz, Bolivia
[4]Department of Atmospheric and Oceanic Sciences, University of Maryland, College Park, College Park, Maryland, USA
[5]Institute of Physics, University of Sao Paulo, Sao Paulo, Brazil
[6]Shanghai Key Laboratory of Atmospheric Particle Pollution and Prevention (LAP3), Department of Environmental Science & Engineering, Fudan University, Shanghai, China
[7]Federal University of Uberlândia, Uberlândia, Brazil
[8]Laboratory of Atmospheric Chemistry, Paul Scherrer Institute, Villigen PSI, Switzerland
[9]Institute of Coastal Systems - Analysis and Modeling, Helmholtz-Zentrum Hereon, Max-Planck-Str. 1, 21502 Geesthacht, Germany
[10]Joint International Research Laboratory of Atmospheric and Earth System Sciences, Nanjing University, Nanjing, China
[11]Aerosol and Haze Laboratory, Beijing Advanced Innovation Center for Soft Matter Science and Engineering, Beijing
University of Chemical Technology, Beijing, China
[12]Institut des Géosciences de l'Environnement, Université Grenoble Alpes, CNRS, INRE, IRD, Grenoble INP, 38000 Grenoble, France
[13]Institute of Atmospheric Sciences and Climate, National Research Council of Italy (CNR-ISAC), 40129 Bologna, Italy
[14]Institute for ion and applied physics, University of Innsbruck, Innsbruck, Austria
[15]Université Clermont Auvergne, CNRS, Laboratoire de Météorologie Physique (LaMP) F-63000 Clermont-Ferrand, France
[16]Leibniz Institute for Tropospheric Research, 04318 Leipzig, Germany
[17]Aerodyne Research, Inc., Billerica, Massachusetts, USA
[18]Institute of Physics, University of Tartu, W. Ostwaldi 1, EE-50411, Tartu, Estonia
[19]Department of Mechanical Engineering, IUPUI, Indianapolis, IN, USA

[a]now at: Department of Chemistry and Molecular Biology, University of Gothenburg, 41296, Gothenburg, Sweden

*Correspondence to*: Victoria A. Sinclair (victoria.sinclair@helsinki.fi)



## Abstract

In this study, we investigate atmospheric new particle formation (NPF) across 65 days in the Bolivian Central Andes at two locations: the mountain-top Chacaltaya station (CHC, 5.2 km above sea level) and an urban site in El Alto-La Paz (EAC), 19 km apart and at 1.1 km lower altitude. We categorize days into four groups based on NPF intensity, determined with the daily maximum concentration of 4-7 nm particles: (A) high at both sites, (B) medium at both, (C) high at EAC but low at CHC, (D) and low at both. This categorization was premised on the assumption that similar NPF intensities imply similar atmospheric processes. Our findings show significant differences across the categories in terms of particle size and volume, precursor gases, aerosol compositions, pollution levels, meteorological conditions, and air mass origins. Specifically, intense NPF events (A) increased Aitken-mode particle concentrations (14-100 nm) significantly on 28% of the days when air masses passed over the Altiplano. At CHC, larger Aitken-mode particle concentrations (40-100 nm) increased from $1.1 \times 10^3$ cm$^{-3}$ (background) to $6.2 \times 10^3$ cm$^{-3}$ very likely linked to the ongoing NPF process. High pollution levels from urban emissions on 24% of the days (B) were found to interrupt particle growth at CHC and diminish nucleation at EAC. Meanwhile, on 14% of the days, high concentrations of sulphate and large particle volumes (C) were observed, correlating with significant influences from air masses originating from the actively degassing Sabancaya Volcano and a depletion of positive 2-4 nm ions at CHC. During these days, reduced NPF intensity was observed at CHC but not at EAC. The study highlights the role of NPF in modifying atmospheric particles and underscores the varying impacts of urban versus mountain-top environments on particle formation processes in the Andean region.



# 1 Introduction

Atmospheric new particle formation (NPF) is a process when under favourable conditions, molecules nucleate to form
particles and clusters and through further condensation and coagulation grow to sizes up to tens of nanometers in diameter.
Several modelling studies have shown that NPF can contribute globally to over 50% of cloud condensation nuclei (CCN)
(Dunne et al., 2016; Gordon et al., 2017; Kerminen et al., 2018; Merikanto et al., 2009; Spracklen et al., 2008). Furthermore,
NPF is crucial from a climate perspective because of its potential to affect, to a large degree, cloud radiative properties, cloud
lifetime, and precipitation formation. Therefore, understanding the processes involved in NPF and its role in the production
of CCN is crucial for a better understanding of the current and future climates as well as air quality and related effects.

In situ measurements are needed for understanding the diverse and heterogenous NPF processes worldwide and they are
crucial for validating theoretical models and assessing the relevance of laboratory experimental results. This is especially
important given the global variation in NPF mechanisms and their climatic and health impacts (e.g. Lehtipalo et al., 2018;
Yao et al., 2018; Daellenbach et al., 2020; Gordon et al., 2017). Hence, conducting strategically well-distributed
measurements globally is essential to ensure a comprehensive understanding of NPF and its implications in the real
atmosphere (Kulmala, 2018)

Despite the growing number of global observations, there is a pronounced bias toward studies conducted in the Northern
Hemisphere and at lower altitudes, leaving the Southern Hemisphere and high-altitude (urban or background) regions
underrepresented in NPF research (Laj et al., 2020; Nieminen et al., 2018). Particularly in South America, research on new
particle formation, especially ground-based studies, is recent and limited, predominantly focusing on the Amazon rainforest
(Varanda Rizzo et al., 2018; Wimmer et al., 2018) or urban settings at low elevations in São Paulo, Brazil (Backman et al.,
2012; Monteiro dos Santos et al., 2021). While there have been numerous airborne studies exploring NPF in relation to
convective cloud processes in the Amazon (Andreae et al., 2018; Krejci et al., 2003, 2005; Williamson et al., 2019; Zhao et
al., 2020, 2022), these investigations, despite providing valuable insights, offer limited long-term continuity due to their
narrow temporal coverage and focus on part of the atmosphere with different thermodynamic conditions and atmospheric
dynamics compared to the Earth surface (Collaud Coen et al., 2018).

In this context, the Global Atmosphere Watch Station at Chacaltaya (hereafter referred to as CHC) emerges as a key site to
close this gap. Positioned at a high altitude (5240 m above sea level) in the Bolivian Andes, at the intersection of the
Amazon basin and the Altiplano plateau (hereafter Altiplano), CHC offers a unique vantage point for NPF studies. At this
site, Rose et al. (2015) documented frequent NPF events based on a year-long study conducted in 2012. They found that both
the formation rate and frequency of NPF events were significantly higher during the dry season, in contrast to the wet
season. Additionally, their examination of air mass trajectories revealed that NPF events are more likely to occur when air





masses originate from the direction of the Pacific Ocean. However, several questions are still unanswered. For example, we need to better understand the chemical composition of the precursor gases involved in nucleation and growth, as well as the composition of the aerosols formed during these events. Furthermore, a more detailed understanding of the atmospheric transport processes in this complex high-altitude environment is necessary, including disentangling how various atmospheric

layers interact and the specific impacts of emissions from nearby La Paz/El Alto conurbation and volcanic activities on the NPF process.

The challenges faced by Rose et al. (2015) in capturing the full vertical and horizontal scope of NPF events and assessing the impact of nearby emissions on CHC are indicative of a widespread difficulty in single-site NPF observations (and especially

at mountain top sites with complex topography). Relying on measurements from just one station assumes uniformity in the characteristics of all sampled air masses, an assumption that is occasionally valid in uniform (regional) NPF. Yet, the atmospheric complexity often restricts the comprehensive representation and understanding of NPF processes observed from a single vantage point. Expanding simultaneous observations at multiple sites within a localised region can help overcome these limitations by enabling the distinction between localized and more regional NPF events and how these affect individual

sites. At mountain top sites this becomes more important due to the strong diurnal cycle in thermally driven winds and planetary boundary layer (PBL) structure in these regions.

Due to the nature of NPF events, which last for several hours and are influenced by the path and history of the involved air masses, any NPF-related observation at a fixed point in space will depend strongly on the origin and path that the air mass

has taken through the atmosphere before arriving at the measurement point. Whether the air mass has passed over polluted urban areas with high emissions or has descended from the clean upper troposphere will influence the chemical and physical properties of that air mass. For this reason, accurately tracing the geographical history of the air mass before the sampling point is crucial.

This is typically done by computing single back trajectories (a.k.a. trajectory models; Fleming et al., 2012) using meteorological data, which is often of coarse resolution from global reanalysis (Fleming et al., 2012 and references therein). However, single-back trajectories do not account for the turbulent and chaotic nature of the atmosphere and lack any probabilistic information on where the air masses originated from. Furthermore, backward trajectories do not account for filamentation and the backward volume growth of the measured air masses (Stohl et al., 2002). To mitigate these

disadvantages, particle dispersion models can be used. Moreover, the results from any dispersion or trajectory models are also sensitive to the quality of the model-based meteorological input data (Foreback et al., 2024) and this is especially true in areas of complex topography, where coarse resolution meteorological data will not resolve small-scale flows such as thermally driven upslope and downslope winds. Therefore, in this study, we analyze air mass origin using the Lagrangian



dispersion model, FLEXPART, which has been driven by high resolution meteorological model (horizontal grid spacing

down to 1 km) data (Aliaga et al., 2021).

Another significant challenge in the study of NPF is identifying days on which NPF events are observed (Kulmala et al., 2024). Traditionally these are classified into event days, non-event days, and undefined days (e.g. Dal Maso et al., 2005; Kulmala et al., 2012; Dada et al., 2018). Event days are characterised by a clear new mode of particles present in the particle

number size distribution (PNSD) with a subsequent growth that can be followed for several hours, while non-event days display an absence of new particles in the nucleation mode size range. Undefined days are those in between that do not fulfil either of these two criteria. This approach, however, faces two key issues: a substantial number of days fall into the undefined category often due to non-stationary weather conditions, which makes statistical analysis complicate; and the traditional classification fails to identify subregional NPF events. Recently, the nanoparticle ranking analysis was introduced

(Aliaga et al., 2023). Unlike the conventional method of day classification, this method assigns values (or rankings) to days based on the concentration of nanoparticles within the smallest measurable size ranges (for example, 2.5–5 nm). This analysis is not only a more nuanced indicator of NPF activity but also correlates well with the formation rate, offering a continuous and robust metric for evaluating the probability and intensity of NPF events on all days (including those previously falling into the undefined group).


The Southern hemisphere high ALTitude Experiment on particle Nucleation And growth (SALTENA, December 2017 to May 2018, Bianchi et al., 2022), aimed to understand the formation and growth mechanisms and properties of aerosols in the Andes. It focused on observation sites at CHC and the La Paz/El Alto conurbation (EAC; Fig. 1). This campaign has provided detailed new information on the transport mechanism of the air masses (Aliaga et al., 2021), the molecular

composition of the nucleating precursors and clusters (Zha et al., 2023a), quantitative information on dimethyl sulphide (DMS) and its oxidation products (Scholz et al., 2022) and the vapours contributing to particle growth (Heitto et al., 2024). Additionally, it documented a significant rise in volcanic degassing in the area (Moreno et al., 2024), and produced a new dataset detailing the PNSD and aerosol chemical compositions at El Alto City (EAC; Bianchi et al., 2022; Mardoñez-Balderrama et al., 2024). However, these findings have yet to be fully incorporated into the current understanding of the NPF

process in the region, and in particular, a comparison between the observations at EAC and CHC has not been explored.

In this study we expand our current understanding of NPF in the central Andes through the analysis of 65 days of observations (March 19 – May 31, 2018) at CHC and EAC that correspond to the transition period from the wet to the dry season. We apply the recently developed "nanoparticle ranking analysis" (Aliaga et al., 2023) to evaluate the probability and

intensity of NPF occurrence simultaneously at both sites, resulting in a joint distribution. This analysis is integrated with an in-depth examination of air mass histories at CHC (Aliaga et al., 2021) and EAC, and complemented with observations of



aerosol chemical composition concentrations (Bianchi et al., 2022), sulfuric acid measurements at CHC (Zha et al., 2023). Our study aims to deepen the understanding of NPF in the Andes, focusing on both background and polluted regimes. We investigate the regional representativeness (both vertical and horizontal) of NPF observations at CHC and their correlation

with events at EAC. We also quantify the impact of anthropogenic emissions at EAC, locally and at a sub-regional level. Moreover, we identify local and regional meteorological patterns and air transport mechanisms that influence NPF, assessing their effects on both sites. Lastly, we evaluate the contrasting influence of volcanic plumes on NPF at CHC and EAC.

# 2  Observations and instrumentation

This study is based on measurements obtained during the Southern hemisphere high ALTitude Experiment on particle

Nucleation And growth (SALTENA) campaign. The experimental setup and overview for this campaign have been previously detailed by Bianchi et al. (2022). The SALTENA campaign spanned from December 2017 to May 2018, aiming to understand the formation and growth mechanisms and properties of aerosols in the Andes. This study focuses on 65 days in the period between March 19 and May 31, 2018 (9 days were excluded due to missing PNSD observations at either site) when intensive measurements were concurrently performed at the sites of CHC and EAC. Subsequent sections below detail

the instrumentation employed for this study.

## 1.1  Site descriptions

### 1.1.1  The Global Atmosphere Watch Chacaltaya Station

The Global Atmosphere Watch Chacaltaya Station (CHC) situated at 5240 m above sea level (a.s.l.) at coordinates 16.350°S, and 68.131°W (Fig. 1), was established in 2011 and has been extensively described in several studies (Adams et al., 1977,

1980, 1983; Aliaga et al., 2021; Andrade et al., 2015; Bianchi et al., 2022; Chauvigné et al., 2019; Francou et al., 2000; Heitto et al., 2024; Koenig et al., 2021; Mardoñez-Balderrama et al., 2024; Moreno et al., 2024; Ramirez et al., 2001; Rose et al., 2015, 2017; Wiedensohler et al., 2018; Zha et al., 2023b, a). Located approximately 140 m below the peak of Mount Chacaltaya and about 1400 m above the Altiplano, CHC is part of the Cordillera Real Mountain range. This range extends from southeast to northwest, forming a natural divide between the Altiplano to the west and the Amazon to the east. The

conurbation of La Paz/El Alto is situated approximately 19 km south of CHC.

CHC experiences an annual cycle of dry and wet seasons. The wet season spans 4 months, from December to March while the dry season spans 4 months from May to August. April is considered as the transition period between wet to dry seasons and September to November the dry to wet transition period. The station records an annual average atmospheric pressure of

534 hPa, temperature of 0°C, and an annual precipitation of 865 mm (Perry et al., 2017). It is predominantly influenced by air masses advected from the west during the dry season and from the lowlands to the northeast and southeast during the wet



season (Chauvigné et al., 2019). Actively degassing volcanoes in the Peruvian Western Cordillera such as Sabancaya (5976 m a.s.l., ~400 km north-west of CHC) routinely influence the observed sulphate concentrations at CHC (Aliaga et al., 2021; Moreno et al., 2024). Pollution from the conurbation of La Paz/El Alto influences observations at CHC regularly,

preferentially during the early afternoon due to PBL growth and orographic thermal winds (Chauvigné et al., 2019; Wiedensohler et al., 2018). In general, at any given time, about a quarter of the air masses arriving at CHC have been in touch with the pseudo PBL (below 1.5 km) within 4 days before arriving at the station (Aliaga et al., 2021).

### 1.1.2    The El Alto City urban background site

EAC is located at the meteorological station of El Alto International Airport. This site has been previously described

(Bianchi et al., 2022; Mardoñez-Balderrama et al., 2024; Wiedensohler et al., 2018) and is situated at an altitude of 4040 m a.s.l., at coordinates 16.510°S and 68.199°W (Fig. 1). It is located on the eastern edge of the Altiplano and at the western high and flat area of the conurbation of La Paz/El Alto (orange region in Fig. 1), which has an estimated population of 2 million inhabitants.

The minimum distance from the meteorological station to the surrounding city's roads and neighbourhoods is ~600m to the south and northwest. The building of the meteorological station is 250 m away from the airport runaway. The airport experiences low traffic, and little influence from planes has been observed. The site is representative of the urban background conditions of El Alto City. Meteorological seasons at EAC closely resemble those at CHC, but EAC, situated at a lower altitude, experiences higher annual mean temperatures and pressures (8°C and 664hPa; Mardoñez-Balderrama et al.,

205    2024).

## 1.2    Size-Resolved Particle Number Concentrations

Size-resolved number concentrations for particles between the electrical mobility diameters of 2 and 440 nm were concurrently measured at CHC and EAC. This diameter range was obtained by combining the measurements of the Neutral Cluster and Air Ion Spectrometer (NAIS) for ions between 2 and 4 nm and particles between 4 and 15 nm and the Mobility

Particle Size Spectrometer (MPSS) for larger particles between 15 nm and 440 nm. In practice, there are overlapping diameters between the instruments, which we used for calibration and validation as described below. The time resolution of both instruments is below 10 minutes.

The NAIS (Mirme and Mirme, 2013) nominally measures the number size distributions of ions in the electric mobility

diameter range between 0.842 nm and 42 nm and particles in the range between 2.5 and 42 nm. The NAIS operates with two parallel measurement columns, each dedicated to a specific polarity. During ion measurements, positive and negative ions are simultaneously measured in their respective columns. For particle measurements, aerosol particles are charged to



opposite polarities using corona chargers and then measured concurrently in their respective columns. It is important to note that particle observations below about 2.5 nm are contaminated by charger ions and therefore are excluded from the

measured size range for particles. The separation of air ions and charged particles is based on their electrical mobilities, with detection carried out by a multichannel differential mobility analyzer (DMA).

At EAC, we utilized the newer NAIS 5 model. This model is designed to adapt to atmospheric pressure changes, mainly because it features two blowers per analyzer that can adjust to varying ambient pressures. In contrast, at CHC, we used the

older NAIS 3 model. This model does not adjust to atmospheric pressure changes and operates with a single blower per analyzer that manages both the sheath and sample flow. The NAIS 3 model at CHC has been previously described (Rose et al., 2015). During our campaign, it was necessary to adjust the raw observations due to the instrument's inability to adapt to the low pressure of CHC and the blower's reduced performance. This resulted in a lower-than-expected flow rate, which in turn led to incorrect sizing of the particles. The adjustment procedure included a simultaneous 5-day measurement with both

NAIS 3 and NAIS 5 at CHC, using the NAIS 5 as a reference for correcting the NAIS 3. Additionally, using the MPSS as a standard, the NAIS 3 was adjusted based on the overlapping size region with the MPSS (10 to 42 nm). This overlap was then used to correct the NAIS 3 measurements. As a result of the above-described sizing issues and to homogenize the NAIS measurement diameter range across sites, we decided to report ions between 2 and 4 nm and particles between 4 nm and 15nm. At these size ranges we have high confidence on the performance of both NAISs.


Two MPSSs (TROPOS-type, Wiedensohler et al., 2012) were also deployed at CHC (permanently) and EAC (during SALTENA). The instrument deployment at CHC has been previously detailed (Wiedensohler et al., 2018), and a similar protocol was followed at EAC. At CHC, the MPSS measures particle sizes ranging from 10 to 460 nm, while at EAC, the size range extends to 692 nm. However, to homogenize the measurements across sites and to combine these measurements

with those of the NAIS, we selected the diameter range between 15 and 440 nm. Weekly calibrations were performed using standard latex particles of 203 nm (Wiedensohler et al., 2012). Additionally, a weekly verification of the aerosol inlet and CPC flow rates was conducted to ensure accuracy.

## 1.3   Mass concentrations of particle chemical composition

Mass concentrations of particle chemical composition were concurrently measured at the CHC and EAC sites. The mass

concentrations of sub-micrometer non-refractory aerosol chemical species, such as sulphate, were determined using a Quadrupole Aerodyne Aerosol Chemical Speciation Monitor (ACSM; Aerodyne Research Inc.; Ng et al., 2011) whereas measurements of equivalent black carbon were conducted using the Aethalometer AE31. The procedures for deploying the AE31 instruments at the stations are documented in Mardoñez-Balderrama et al. (2024). Similarly, the methodology for ACSM deployment at CHC, which was identical to that at EAC, is outlined in Aliaga et al. (2021).






The ACSM is used for characterizing non-refractory submicron aerosol species, including organics, nitrate, sulphate, ammonium, and chloride (Ng et al., 2011). Due to the challenges posed by low atmospheric pressure, a modification was made to the ACSM by incorporating a critical orifice with a diameter of 130 μm. This adaptation was essential to maintain the standard sample mass flow rate (Fröhlich et al., 2013). After passing the critical orifice, the air sample travels through an

aerodynamic lens, which enables particles between ~75–650 nm in vacuum aerodynamic diameter to reach a particle vaporizer (Liu et al., 2007). Here, particles are flash vaporized at 600 °C and ionized by electron impact ionization (70 eV). The particle composition is thereafter determined using quadrupole mass spectrometry. To ensure optimal instrument performance and accurate mass quantification, calibrations were routinely conducted during the campaign. These calibrations focused on the instrument's inlet flow and mass, using ammonium nitrate and ammonium sulphate as standard

references. The ACSM was set to record data at a temporal resolution of 30 minutes and a $PM_{10}$ cyclone was used in the line as cut-off for the aerosol particle measurements. In addition, the Magee Nafion membrane dryer was used to guarantee aerosol particles below 50% relative humidity.

During some volcanic plume events at CHC, the ACSM inlet became clogged, resulting in underestimated concentrations for

all measured species. To correct this, we analyzed the ratio of observed volume from the PNSD and mass concentration from the ACSM between 0:00 and 4:00— a period when eBC is negligible and does not skew the ratio. This ratio remained stable when the instrument was unclogged. Thus, during the clogged periods, we adjusted the species concentrations upward to match the ratio observed on unclogged days.

The AE31 (Magee Scientific, Arnott et al., 2005, Tape: Pall flex Quartz fiber Q250F) is a filter-based absorption photometer that measures light attenuation by atmospheric aerosol particles at 7 wavelengths in the visible/near-visible spectrum (370, 470, 520, 590, 660, 880, 950 nm). Measurements at 880 nm were used to obtain an estimate of the equivalent black carbon mass concentration (Petzold et al., 2013).

## 1.4    Ancillary measurements

Routine meteorological parameters (e.g., temperature, relative humidity, and wind speed and direction) were measured continuously at both CHC and EAC (Bianchi et al., 2022) at varying temporal resolutions and subsequently averaged to a uniform 15 minute temporal resolution. Additionally, incident global shortwave radiation (hereafter referred to as "incident SW radiation") measurements were taken at CHC but not at EAC.

We also measured sulfuric acid (SA) concentrations at CHC with the chemical-ionization atmospheric-pressure-interface time-of-flight (CI-APi-TOF; Aerodyne Research Inc. & Tofwerk AG) mass spectrometer (Jokinen et al., 2012). Zha et al.



(2023) provide comprehensive installation and operational details of this instrument's deployment at the station. The instrument was a Nitrate ion (NO₃)-based CI-APi-TOF, widely used for atmospheric SA measurements (Bianchi et al., 2016; Jokinen et al., 2012).


Cloud cover fraction (CF, Fig. S6) data for the stations is gathered from the MODIS instrument aboard the Aqua and Terra satellites, which pass over the region four times daily at ~10:00, ~14:00, ~20:00, and ~02:00 local time (Platnick et al., 2015) and have a spatial resolution of approximately 5 km. Due to higher noise levels in night-time data, we focus on daytime retrievals. For each satellite pass and day, we calculate the CF at each station by averaging the cloud fraction values from all

pixels within a 5 km radius centred on each station.

# 3    Methods and diagnostics

## 1.5    Categorization of days based on NPF intensity across both sites

Traditional atmospheric NPF field studies have relied on predefined criteria to differentiate between days with and without NPF (e.g. Dal Maso et al., 2005; Kulmala et al., 2012; Dada et al., 2018), but this binary approach often misclassifies weak

events and non-regional phenomena, limiting further analyses such as growth and formation rates. Kulmala et al. (2024) proposed deriving a probability distribution for NPF intensity across all observed days, using nanoparticle ranking analysis (Aliaga et al., 2023) to produce a daily metric indicative of NPF intensity. This metric, which focuses on the daily maximum minus background particle concentration within the nucleation mode, correlates higher values with increased formation rates and event probabilities. Additionally, grouping days by shared characteristics, such as chemical environment, season, or air

mass origin, as demonstrated by Kulmala et al. (2022, 2024), helps identify emerging patterns and mitigates observational noise, leading to more accurate estimations of metrics like GR.

Building upon these recent advancements, and the inclusion of two observational sites, this study employs a straightforward, yet effective methodology based on the nanoparticle ranking analysis (Aliaga et al., 2023) and k-means clustering to create

four categories of days for further analysis. First, for each site, we derive a single metric for daily NPF intensity using the diameter range of 4 to 7 nm for the number concentration ($N_{4-7}^{\mathrm{day\,max}}$). For the $N_{4-7}^{\mathrm{day\,max}}$ calculation, following Aliaga et al. (2023), we identify the maximum concentration of $N_{4-7}$ during the active period (8:00 to 18:00) after smoothing the $N_{4-7}$ time series with a two-hour rolling median. Diverging from Aliaga et al. (2023), we select a 4-7 nm diameter range (as mentioned above), considering our confidence in particle measurement reliability above 4 nm at CHC and aiming for a range

that ensures a detectable signal while diminishing influence of potential primary emissions. Instead of calculating the difference between $N_{4-7}^{\mathrm{day\,max}}$ and background median, as suggested by Aliaga et al. (2023), we opt for a simpler method of using only the maximum concentration during the active period, having found both approaches yield comparable results.





We then construct a two-dimensional vector for each day, with $N_{4-7}^{\text{day max}}$ from both sites. Utilizing k-means clustering, we

cluster days into four categories based on the NPF intensity at the two sites, anticipating two categories with uniformly high or low intensities across both sites and two with mixed intensities (Fig. 3). Furthermore, we also employed three popular metrics to identify the optimal number of clusters: Silhouette Scores, Davies-Bouldin Index, and Calinski-Harabasz Index. Among configurations ranging from two to six clusters, the four-cluster option achieved the highest overall score across these tests. The category separation not only produced significantly distinct values of $N_{4-7}^{\text{day max}}$ for each category but also

produced significantly (Mann-Whitney U test, p-value<0.1) distinct values for, among others, $J_{4-7}$, PNSD, aerosol chemical compositions (e.g. eBC, Sulphate), SA and meteorological parameters (e.g. RH, CF, WVMR; Figs. S13-S16)

Finally, we also used a more traditional approach to classify days loosely based on Dada et al. (2018) to compare with our categorization scheme (Figs. S4 and S8). We utilized concentrations of negative ions in the 2-4 nm range and particle

concentrations from 7 to 25 nm. We conducted a visual inspection on the PNSD for each day at each site, identifying days with an observed increase in $N_{2-4}^-$ lasting more than one hour as nucleating days (Nuc-D). Days showing a noticeable growing edge in the range from 4 to 25 nm were labelled as Growth days (Gr-D). Additionally, days with a distinct increase in the 7-25 nm range were designated as Aitken peak days (Ait-P-D). It is important to point out that at EAC, many days were classified as Nuc-D and Ait-P-D but not Gr-D since likely the local dynamics do not allow the growing edge to be

observed.

## 1.6  Air mass origin analysis

All analyses of air mass history in this study are derived from 4-day backward simulations using the Lagrangian FLEXible PARTicle (FLEXPART; version FLEXPART-WRF_v3.3.2; Brioude et al., 2013) dispersion model and using CHC and EAC as the arrival points (details in the following paragraphs). For CHC, we utilized a dataset previously generated by (Aliaga et

al., 2021), while for EAC, new FLEXPART simulations were run with an identical setup but targeted EAC as the arrival point. We refined the results from Aliaga et al. (2021), which used a pseudo PBL (defined as less than 1.5 km), by employing a more precise PBL metric. We determined whether air masses were within or above the PBL by comparing their geographic positions and altitudes against the PBL height in the meteorological dataset used for the simulations.

Aliaga et al. (2021), produced a high-resolution (up to 1 km) meteorological data set for a large area around CHC and for the 6-month duration (2017-12-06 to 2018-05-31) of the SALTENA campaign by running the Weather Research and Forecasting (WRF) model version 4.0.3 (Skamarock et al., 2019), which is a state-of-the-art, non-hydrostatic, regional numerical weather prediction model. The initial and boundary conditions were taken from the National Centers for Environmental Prediction Climate Forecast System Version 2 (Saha et al., 2011, 2014). Four nested domains were included, with the outermost domain





covering large parts of South America and the Pacific Ocean with a grid spacing of 38 km and the innermost domain covering an area of approximately 180 km by 180 km centered on CHC with a grid spacing of 1 km. The model output was saved every 15 minutes. Further details of the WRF model simulation set up are given in Aliaga et al. (2021).

Aliaga et al. (2021) also ran FLEXPART simulations to identify the origins of air masses arriving at CHC. For the
campaign's duration, 20,000 virtual particles (virtual passive air tracers) were released every hour and tracked backwards for four days before their arrival at CHC. In this study, we utilize this pre-existing dataset and additionally, we also performed new FLEXPART simulations to identify the origins of air masses arriving at EAC. These additional FLEXPART simulations were also driven by the WRF model output from (Aliaga et al., 2021) and again 20,000 virtual particles were released every hour.


When run in backwards mode, FLEXPART computes the emission sensitivity response function, also referred to as the source–receptor relationship (SRR), on a user-specified three-dimensional longitude–latitude–height grid (Eckhardt et al., 2017; Pisso et al., 2019; Seibert and Frank, 2004). The potential emission sensitivity provides a footprint of emission source areas and if multiplied by actual emissions would give an estimate of the concentrations that would be measured at the
receptor (station). The SRR output was casted into a log polar grid centered at each of the stations following the methods and rationale described by Aliaga et al. (2021)

In the results section, we distinguish air mass history in the long-range, medium-range and short-range. The long-range transport is analyzed by the integrated SRR over the full 4-day duration of the simulated transport back in time. The medium
range is calculated in the same way as the long-range but with a color bar scaled to emphasize sources within a 200 km range. The short-range transport is calculated slightly differently for a more detailed representation of the air mass movement close to the station: For 1, 2, 3, and 4 hours prior to arrival at the site, the SRR at the time was clustered into 20 clusters (using k-means clustering), forcing half of the clusters to be above and half of them to be below the boundary layer. The mean location of each cluster was then calculated in addition to the size of the SRR magnitude in the cluster. In total each
cluster is characterized as a "blob" with a certain location and a volume representing the SRR contribution of this cluster. Since 20 blobs are produced per hour back, the full transport 4 hours prior to arrival at the station is represented by a total of 20×4= 80 blobs.

## 1.7   Formation rate, growth rate, coagulation sink, and condensation sink calculations

Particle formation rates ($J$) and the growth rates (GR) are the most important parameters characterizing atmospheric NPF
(Kerminen et al., 2018). Additionally, coagulation sink (CoagS) and condensation sink (CS) are commonly used when analysing NPF processes. These parameters can be calculated in a microscopic or macroscopic fashion but in field studies



the latter is usually applied because then these quantities can be directly estimated from the measured particle size distribution.

The formation rate is defined as the flux of growing particles above a threshold diameter (in our case 4 nm) and is denoted as $J_4$ and its units are (particles) cm$^{-3}$ s$^{-1}$. It can be calculated from the measured particle size distribution following the formulation based on the aerosol general dynamic equation (Kulmala et al., 2012),

$$J_4 = \frac{\Delta N_{[4,7)}}{\Delta t} + \text{CoagS}_{[4,7)} \times N_{[4,7)} + \frac{\text{GR}}{(7-4)\text{nm}} \times N_{[4,7)} \qquad (1),$$


where $N_{[4,7)}$ is the concentration of particles sized from 4 nm up to, but not including, 7 nm.

The growth rate (GR) indicates the time-based change in diameter (D) of the growing mode in the PNSD. GR can be estimated in field studies using the mode-fitting, maximum-concentration or appearance time methods (Kulmala et al., 2012;
Lehtipalo et al., 2014). The mode-fitting method fits a log-normal mode to the particle size distribution at each time step of a formation event and uses the geometric mean diameter for the growing mode, with GR calculated from the slope of a least-squares line through these means. The maximum concentration method identifies peak concentrations at each size bin after applying a Gaussian filter for noise reduction and fits a least-squares line to these peaks with swapped axes (x-axis is diameter, y-axis is time) since the uncertainty is in the time dimension, calculating GR as the inverse of this slope. The
appearance time method is similar to the maximum concentration method but instead of using the maximum, a certain threshold of the maxima (e.g. 50%) is used. In this study we use the maximum concentration method (Max) and the appearance time method setting the threshold as the maximum of the time derivative (Der.). We do not use the mode-fitting method because events in this region nucleate for many hours making the method unsuitable for this environment. Results from the Max and Der. methods are shown in Figs. S1 and S2, and Tables S1 and S2 but we relied on the Der.
method for $J_4$ estimations (Eq. 1) because it better handles the increasing surface influence at CHC. Mathematically, it is precise to use the GR at the 7 nm upper size limit for Eq. (1), but commonly any nucleation mode GR is used. We used GR in the range from 4 to 7 nm.

The coagulation sink, calculated as

$$\text{CoagS}_{[4,7)} = \sum_{i=j}^{i=n} K\left(\sqrt{4 \cdot 7}\text{ nm}, \sqrt{D_{L,i} \cdot D_{U,i}}\right) \cdot N_{[D_{L,i},D_{U,i})} \quad (2),$$

quantifies the rate at which particles within a specific size range, here [4,7) nm, collide and coalesce with particles of similar or larger sizes, thereby exiting the growing particle mode. The variables $D_{L,i}$ and $D_{U,i}$ denote the lower and upper diameter



boundaries of each diameter size bin $i$. The integer $j$ is the number of the first bin for which the lower limit of the bin is larger than 4 nm ($j = \min\{i \text{ s.t. } D_i > 4 \text{ nm }\}$). The integer $n$ equals the total number of bins and $D_{U,n}$ the upper diameter of the last bin which in our case is 470 nm. Each term in the sum involves a coagulation coefficient $K$ which takes two parameters: the first parameter is the geometric mean between 4 and 7 nm and the second parameter is the geometric mean between the diameter boundaries of bin $i$. The coagulation coefficient is multiplied by the number concentration of particles $N$ in bin $i$.

The condensation sink quantifies the theoretical loss rate of condensing vapor molecules (here defined with respect to sulfuric acid) to the existing particle population. Similarly to the coagulation sink, it can be calculated from the particle number size distributions using the equation (Dal Maso et al., 2002)

$$\text{CS} = 4\pi D^* \sum_{i=1}^{i=n} \beta\left(\sqrt{D_{L,i} \cdot D_{U,i}}\right) \cdot N_{[D_{L,i},D_{U,i})} \qquad (3)$$

where $D^*$ is the diffusion coefficient of sulfuric acid, and $\beta$ is the transition regime correction factor.

In field measurements, the most difficult quantity to estimate is the GR as it can only be calculated when a growing mode is observed and additionally it can be affected by the fluctuating air masses and/or primary emissions. Moreover, a changing environment upwind can even produce apparent shrinking (Hakala et al., 2023). As mentioned above, Kulmala et al., (2022) have suggested to calculate GR by taking group averages of similar days. In this study we follow this approach by using the median GR₄₋₇ of all the days within a category where GR₄₋₇ was available. Then this category median GR₄₋₇ is used in Eq. (1) together with the other required quantities.

# 4    Results and Discussion

In this section, we describe the results of our analysis and discuss their implications. We have clustered the 65 days under study into four categories (each day belongs to only one category and 9 days were excluded due to missing $N_{4-7}$ measurements at either location). The classification of each day was based only on the combined 2-dimensional (CHC and EAC) maximum daily values of $N_{4-7}$ ($N_{4-7}^{\text{day max}}$; see Fig. 3 and description in Section 3.1)

The underlying assumption behind this categorization is as follows: The atmospheric processes behind days with similar NPF intensities ($N_{4-7}^{\text{day max}}$) are likely similar to each other and different from days with different NPF intensities. Dividing



these days into a small number of categories allows us to analyse these categories in detail and thus understand which specific processes lead to different NPF intensities. Conducting this categorization at two points that are close in geographical distance but separated by altitude and local influences also allows us to understand which processes are favourable at each specific site.

We first tested whether the four categories were significantly different from each other using 25 selected variables (Figs. S13-S16) including concentrations of particles at four different sizes, total concentration and volume, meteorological parameters (RH, WVMR, incident SW radiation, wind speed and direction), chemical composition of aerosols, CS, and SA. For each variable, we collated hourly measurements by category and compared these across all four categories at the same hourly intervals using the Mann-Whitney-U test, which is suited for non-normal distributions (Figs. S13-S16). We
established our null hypothesis that the collated hourly measurements are not significantly different, setting a significance threshold at p-values greater than 0.1. The alternative hypothesis, that collated hourly measurements are distinct, was set for p-values less than 0.1. The findings from this analysis revealed that approximately 48% of the measurement sets exhibited statistically significant differences. In other words, on average each variable is significantly different half of the time. Ideally, a flawless categorization would result in 100% significant differences, indicating clear separations between
categories. In contrast, poor or arbitrary categorization would likely result in values close to 0%, suggesting no discernible differences between categories based on the data. Thus, we concluded that the categories are indeed significantly different in terms of concentrations of size-resolved particle number and volume, precursor gases, aerosol chemical composition, as well as levels of pollution, meteorological parameters, transport patterns and air mass origins (Figs. 8, S13-S16). Additionally, the analysis of the origin of the air masses further supported distinct categories originating from different source regions (Fig. 8).

Subsequently, we identified four key emergent and memorable indicators for naming these categories (Fig. 2, Table 1):

- *Intense-NPF* category because combined $J_4$ is considerably higher at both locations than the other three categories.
- *Cloudy* category given that CF at ~14:00 (coinciding with satellite observations over the region) is significantly higher than the other three categories.
- *Polluted* category because eBC concentrations at 12:00 are significantly elevated at both locations.
- *Volcanic* category given that the daily median Sulphate concentrations are significantly higher, highly likely as the result of the influence from Sabancaya Volcano.

Each parameter exhibited high values in one of the four categories, indicating their dominant influence at CHC and EAC (except for $J_4$ where Intense-NPF and Volcanic are similar *only* at EAC). Hard categorizations may not always align
perfectly with observed features. For instance, certain days categorized as Intense-NPF may also exhibit pronounced eBC peaks at noon. Nevertheless, such occurrences are not the norm, as the value distributions across categories remain notably and significantly dissimilar.





The analysis in this study is primarily based on the median daily patterns (rounded to the hour) of size-resolved particle
distributions, aerosol chemical compositions, comprehensive air mass history analyses (i.e. short-range-<4 h- to long-range-4
days-), meteorological variables (e.g. temperature, RH, wind speed and direction, solar radiation), sulfuric acid
concentrations, and crucial NPF characteristics: formation and growth rates. We further examine particle number
concentrations ($N$) or negative (positive) ions ($N^{-(+)}$) within size ranges determined by specific criteria:

- The 2-4 nm (respective number concentration denoted as $N_{2-4}^{-(+)}$) range primarily observes reliable negative
(positive) ion concentrations, indicating early clustering and growth and the onset of nucleation.
- The range 4-7 nm ($N_{4-7}$) falls within the lower nucleation range, sensitive to the intensity of NPF, marking early
nucleation and growth stages.
- The 7-13 nm interval ($N_{7-13}$) represents the upper nucleation range, reflecting further growth, and is not influenced
by primary emissions at EAC during morning and afternoon rush hours.
- The range 13-40 nm ($N_{13-40}$) corresponds to the lower Aitken mode, where most NPF-produced particles are found
by the end of daily NPF events. This range also captures traffic primary emissions at EAC.
- The 40-100 nm interval ($N_{40-100}$) pertains to the upper Aitken mode, with a small yet significant portion of NPF-
born particles reaching this size. It is also sensitive to traffic primary emissions at EAC.
- The range 100-440 nm ($N_{100-440}$) denotes the accumulation mode, affected by primary emissions from pollution,
dust, and notably, volcanic plume influences in this specific geographic area.

In all figures time references are in local time (UTC-4), with data shown as daily hourly medians, left-aligned, and shaded
for the interquartile range.

In Fig. 3, we start with the days within the category denominated *Intense-NPF* (top right quadrant). During these days, high
intensity NPF (i.e. high values of $N_{4-7}^{\text{day max}}$) is observed at both CHC and EAC and a distinct and intense banana-like shape is
observed in the growing NPF mode. Both CHC and EAC show a clear and intense formation event with a well-defined
growth edge. However, there are two clear differences between CHC and EAC. First, the maximum value of $N_{4-7}$ is fivefold
higher at EAC (48.8×10³ cm⁻³) than at CHC (8.2×10³ cm⁻³). Second, the event at EAC is preceded by the appearance of a
mode with high number concentrations in the Aitken range (13-100 nm; 40.9×10³ cm⁻³) observed from 5:00 to 8:00. This
latter pattern is qualitatively similar in all categories and linked to early morning rush hour and a shallow PBL.

In *Polluted* (Fig. 3, top left quadrant) at CHC we observe morning (9:00-12:00) nucleation and growth similar to what was
observed at Intense-NPF-CHC. However, the event is abruptly interrupted at noon when aerosol concentrations in the
accumulation mode ($N_{100-440}$) rapidly increase from 0.51×10³ cm⁻³ at 10:00 to 1.72×10³ cm⁻³ at 12:00. Later, we will show in
detail that this peak is the result of air pollution advected from EAC and certainly not the result of NPF. At EAC, like





Intense-NPF-EAC, early morning (7:00; $N_{13\text{-}100}$=25.3×10$^3$ cm$^{-3}$) and evening (20:00; $N_{13\text{-}100}$=22.5×10$^3$ cm$^{-3}$) Aitken mode peaks are observed. However, unlike Intense-NPF-EAC, nucleation without apparent growth, is observed here ($N_{4\text{-}7}$ =9.6×10$^3$ cm$^{-3}$ at 13:00, Fig. 3.b and d).

In *Volcanic* (Fig. 3, bottom right quadrant) we observe elevated levels of sulphate and total aerosol volume at both stations. The air masses come from areas near the Sabancaya Volcano (Fig. 1) where active degassing has been observed during the measurement period (Aliaga et al., 2021; Bianchi et al., 2022). Interestingly, at CHC, we see an NPF process with noticeable growth, but at considerably lower intensity ($N_{4-7}^{\text{day max}}$= 1.2×10$^3$ cm$^{-3}$) compared to that of Intense-NPF-CHC ($N_{4-7}^{\text{day max}}$=8.2×10$^3$ cm$^{-3}$). This diminished NPF intensity, however, is not seen at EAC where we observe a similar PNSD

diurnal pattern to that of Intense-NPF-EAC with similar NPF intensity ($N_{4-7}^{\text{day max}}$ is 42.3×10$^3$ and 48.8×10$^3$ cm$^{-3}$ at Volcanic-EAC and Intense-NPF-EAC, respectively). Finally in the lower Aitken range (13–40 nm) the particle number concentration in the afternoon (around 15:00) is comparatively lower than in Intense-NPF-EAC (14.6×10$^3$ vs. 24.2×10$^3$ cm$^{-3}$ for Volcanic-EAC and Intense-NPF-EAC, respectively).

The *Cloudy* category (Fig. 3, bottom left quadrant) is characterized by cloudy skies both at CHC (CF=97% at 14:00, Fig. S6) and EAC (CF=83% at 14:00), the highest relative humidity and air masses originating from the lowlands and valleys east from the Altiplano. At CHC, apparent nucleation is rarely observed during these days (only in seven days out of 22). The overall particle concentration remains exceptionally low with a maximum $N_{3\text{-}440}$ of 3.8×10$^3$ cm$^{-3}$. At EAC, a similar pattern to Polluted-EAC is observed, which is expected as their NPF intensity values are similar ($N_{4-7}^{\text{day max}}$=7.6×10$^3$ cm$^{-3}$ vs 9.6×10$^3$

cm$^{-3}$ for Cloudy-EAC and Polluted-EAC, respectively) and more importantly local anthropogenic sources dominate despite advection from the lowlands.

### 1.7.1    Intense-NPF category

We begin with an overview of the *Intense-NPF* category, initially outlining typical characteristics shared by both CHC and EAC. This is followed by a detailed examination of the median diurnal patterns for each location, focusing on four critical

times: sunrise (7:00), the beginning of new particle formation (NPF) around 9:00, noon (12:00), and sunset (18:00). We conclude by highlighting the key factors associated with NPF and conducting a comparative analysis between CHC and EAC.

During the 18 days within this category, the FLEXPART air mass history analysis (both at CHC and EAC) reveals the

presence of air masses originating from the western Altiplano region. These air mass origins are observed in both the long-range (>200 km; Fig. 4.p), and the medium-range analysis (<200 km; Fig. 8). Generally, these days exhibit cloudless sky conditions (Fig. 4.a) and limited presence of clouds in the region (Fig. S6). Additionally, both sites, when compared to the



other categories, experience the lowest relative humidity levels (<45%; Fig. S9) throughout the day. Within this category a clear and intense process of nucleation (Nuc-D) and growth (Gr-D) has been identified in our conventional classification for all 18 days at CHC and EAC except for two instances at EAC where Gr-D was not detected (Figs. S3 and S4).

At *CHC*, during the pre-dawn period (00:00–06:00), most observed parameters have low values and are either constant in time or decrease gradually. Minimum values of wind speed (1.9 m s$^{-1}$), temperature (-3°C), condensation sink (CS, 2.95×10$^{-3}$ s$^{-1}$), and total particle concentration ($N_{3-440}$, 4.8×10$^3$ cm$^{-3}$) occur at approximately 06:00. Sunrise occurs at ~07:00 after which incident SW radiation and sulfuric acid (SA), which were close to zero during the night, begin to increase. Nucleation begins at ~09:00 which coincides almost exactly with the time when wind speed starts to increase (Fig. 4).

During the early new particle formation phase (09:00—11:00), $J_4$ increases rapidly from 0.4 to 3.4 cm$^{-3}$s$^{-1}$, the CS remains small (0.34×10$^{-3}$ s$^{-1}$) and a GR$_{4-7}$ of 7.0 nm h$^{-1}$ (Table 1) is estimated. Equivalent black carbon (eBC, Fig. 4.c), which is used as a proxy for anthropogenic pollution, also starts to increase during this time interval and the wind speed continues to increase. SA reaches its maximum value (1.35×10$^7$ molecules cm$^{-3}$) at 10:00 after which it starts to decrease, most likely due to the increase in CS and associated scavenging, which reaches 7.5×10$^{-3}$s$^{-1}$ by 12:00. eBC (Fig. 4.c) reaches its peak value of 0.2 µg m$^{-3}$ at 12:00.

In the afternoon (12:00-17:00), the wind is blowing upslope from the southwest and reaches a maximum wind speed of 5.4 m s$^{-1}$ at 16:00. $J_4$ peaks at 14:00 (7.7 cm$^{-3}$ s$^{-1}$) while $N_{13-40}$ and $N_{40-100}$ peak at 14:00 (30.5×10$^3$ cm$^{-3}$) and 16:00 (6.18×10$^3$ cm$^{-3}$) respectively. The peak in the particle concentration in the lower Aitken range ($N_{13-40}$) is likely dominated by the observed NPF process while the peak in $N_{40-100}$ is likely a combination of surfaced-based primary emissions and NPF. In the evening interval (18:00 to 23:00), incident SW radiation, wind speed, and temperature experience a rapid decrease coinciding with sunset. SA and J4 decrease to very low values after 18:00. Meanwhile, eBC and CS decrease more gradually, and their concentrations by the end of the day are only half of their daily maximum.

At *EAC* during the pre-dawn hours (0:00-6:00), the temperature follows a decreasing trend like that seen at CHC, reaching its lowest point at 6:00 (-0.6°C). Wind direction stays consistently north-easterly, with a median speed of 2.1 m s$^{-1}$. The concentration of eBC reaches its lowest at 2:00 (0.4 µg m$^{-3}$) before trending upwards. CS follows a similar trajectory, with its minimum at 4:00 (8.7×10$^{-3}$ s$^{-1}$).

During the early morning hours (7:00-8:00), eBC concentration peaks at 7:00 (3.1 µg m$^{-3}$), a surge attributed to the combined effects of morning traffic, low temperature and a stable, shallow PBL (Wiedensohler et al., 2018). This peak in eBC coincides with morning highs in other parameters: CS at 31.0×10$^{-3}$ s$^{-1}$, $N_{13-40}$ at 26.7×10$^3$ cm$^{-3}$, $N_{40-100}$ at 14.0×10$^3$ cm$^{-3}$, and



$N_{100-440}$ at $3.43\times10^3$ cm$^{-3}$. Likely all of these highs are the result of traffic pollution. Wind speed hits its daily minimum at 8:00 (1.5 m s$^{-1}$), changing direction from north-easterly to westerly and rapidly increasing in speed thereafter. This low wind speed is also seen in the short-range air mass analysis (<4h, Fig. 4.q, grey spheres).

During the early phase of new particle formation (9:00-11:00) at EAC, nucleation is observed around 9:00, leading to a sharp increase in the concentrations of smaller particles: $N_{4-7}$ reaches $48.8\times10^3$ cm$^{-3}$, $N_{7-13}$ $39.9\times10^3$ cm$^{-3}$, and $J_4$ 50.7 cm$^{-3}$ s$^{-1}$ (~ten times higher that $J_4$ at CHC), all peaking at 10:00. In contrast, eBC decreases tenfold to 0.39 µg m$^{-3}$, the CS drops to $11.1\times10^{-3}$ s$^{-1}$, and particle concentrations $N_{40-100}$ and $N_{100-440}$ fall to $3.25\times10^3$ cm$^{-3}$ and $1.15\times10^3$ cm$^{-3}$, respectively, reaching these levels by noon and then stabilizing. The rather swift change in the PNSD, concentration of eBC and value of CS is

likely the result of a rapidly growing PBL. The rapid increase in PBL height is related to the dry environment (RH=36%) which means the latent heat is small and sensible heat flux, which drives the PBL growth, is large.

In the early afternoon (12:00-17:00), at EAC the wind is westerly, and the wind speed stays high at around 3.8 m s$^{-1}$, while temperature peaks at 15:00 (15.3°C). It can be seen that the air masses in the short-range (<4h) come from the Altiplano,

from what appears to be a well-developed PBL that is indistinguishable (in the sense that it follows a very similar path) from the air masses that reach CHC at the same time (Fig. 4.r). The concentration $N_{40-100}$ reaches a minimum of $3.25\times10^3$ cm$^{-3}$ similar to CHC ($3.34\times10^3$ cm$^{-3}$) at the same time which is further evidence that similar air masses are influencing the stations. The concentration of $N_{13-40}$ peaks at 14:00 ($30.9\times10^3$ cm$^{-3}$) most likely as a product of the ongoing NPF process.

In the evening interval (18:00 to 23:00), at EAC there is a marked shift in wind direction from westerlies (from the Altiplano) to easterlies (from the La Paz Valley), which then stabilizes for the rest of the night as the speed decreases (Fig. 4.n). The concentration of eBC and the CS begin to rise again peaking at around 20:00 (2.3 µg m$^{-3}$ and 23.1 $\times10^{-3}$ s$^{-1}$ respectively), likely due to the afternoon increase in traffic-related emissions and the development of a shallow nocturnal PBL. This increase is also clear in the PNSD (Fig. 4.j) where a night-time Aitken mode (centred around 30 nm) is stablished,

similar to that observed in the morning.

#### 1.7.1.1    Similarities and differences

Comparing both locations, we now highlight some key differences and similarities between NPF and other aerosol related processes at CHC and EAC. First, the occurrence of morning and night-time high concentration peaks in Aitken mode in the PNSD is a unique feature at EAC and is absent at CHC. This phenomenon can be linked to the clear decoupling of air

masses during the night, with CHC sampling within the residual layer or free troposphere, while EAC is affected by a shallow nocturnal PBL and advection of urban valley air from La Paz. The presence of these modes at EAC, and their absence at CHC, highlights the impact of local emissions, particularly from traffic, at EAC.





Secondly, $N_{4\text{-}7}$ is significantly higher at EAC, especially during the day (e.g., $45.8\times10^3$ vs $5.9\times10^3$ cm$^{-3}$ at 12:00).

This is likely due to local emissions of precursors at EAC that undergo quick photooxidation during daytime (their concentrations during the early morning rush hour only reaches one-tenth of the daily value). Short range air mass analysis (<4h) suggests similar air mass history at both sites during the day. However, this similarity is partial; closer inspection reveals air masses separate at shorter intervals (<2 h) and in general the residence time of the air masses within the urban area is less than 2 hours. This supports the idea that the higher concentrations of $N_{4\text{-}7}$ at EAC result from local emissions of

precursor gases the lead to higher production of particles.

Thirdly, negative ions $N^-_{2-4}$ persist at EAC until just before 18:00, whereas at CHC, they decrease significantly after 12:00. This persistence at EAC is likely due to the continuous availability of precursor gases from local anthropogenic emissions, while at CHC such emissions are absent, and any precursor gases must be transported, getting lost due the condensation sink

generated by background aerosol concentrations and newly formed particles.

Despite these differences, the aerosol growth patterns are similar after 10:00 (GR$_{7\text{-}13}$ is 7.2 and 7.6 nm h$^{-1}$ for CHC and EAC respectively; Table S1 and Figs. S1 and S2), and the evolution of concentrations of particles $N_{13\text{-}40}$ show great resemblance between the two stations after 12:00. This suggests that during intense-NPF days we encounter regional scale events. The

orographic upslope winds and a well-developed PBL over the Altiplano connect both sites. This hypothesis is supported by almost identical concentrations of $N_{13\text{-}40}$ between 12:00 and 18:00 at both sites (Fig. S17), coinciding with the estimated time it would take the NPF mode to grow to this size. Differences in particle sizes below 14 nm between the sites are likely due to varying anthropogenic influences, particularly from traffic emissions of precursor gases at EAC which increase local production of these smaller particles. In contrast, CHC shows fewer such influences. At EAC, the urban air mass has a short

residence time of approximately 1-2 hours, after which fresher air masses from outside the urban area replace it. These new air masses, unaffected by local emissions, are influenced mainly by regional NPF processes and thus similar to those concurrently observed at CHC.

### 1.7.2 Polluted category

The Polluted category consists of a total of 16 days. At both sites, the FLEXPART air mass history analysis (Fig. 5.p)

reveals that a majority of the airmasses during these days originate from the southeast along the altiplano eastern edge with a lesser contribution from the western direction (Central Altiplano). This spatial origin is evident in both medium-range distances (<200 km) and long-range distances (>200 km, Fig. 8). During these days, the mornings exhibit clear skies; however, as the day progresses, intermittent cloud cover becomes noticeable. This is demonstrated by the increased shaded inter-quantile range in the incident SW radiation curve (Fig. 5.a) and in the increase in the CF fraction (Fig. S6 and S9) from



6% at 10:00 to 15% at 14:00. The relative humidity on these days is higher compared to Intense-NPF, with the median RH
      reaching 48 % at CHC but drier in comparison to Volcanic and Cloudy (Fig. S9).

      Within this category, nucleation is observed on 15 and 14 days at CHC and EAC respectively, while clear particle growth is
      observed on 16 days at CHC and 6 days at EAC (Fig. S4). Moreover, the arrival of a pollution plume around noon (12:00) is
observed at CHC, which frequently interrupts the observed growth process. The arrival of this plume is evident in the
      concentration increase observed in the PNSD surface plot at CHC (Fig. 5.b) around 12:00, seen by an upper range Aitken
      mode centred at ~70 nm in diameter. Additionally, the concentration of eBC shows a substantial increase, with
      concentrations rising tenfold from 0.07 µg m⁻³ at 9:00 to 0.65 µg m⁻³ at 12:00 (, Fig. 5.c).

At *CHC*, in the pre-dawn and early morning period (0:00-11:00), the diurnal pattern of Polluted-CHC is very similar to that
      of Intense-NPF-CHC (Fig. 4). The main difference is that the wind has a more southerly component in this case (Fig. 5.g and
      q, red spheres), the maximum SA concentration is halved (6.05×10⁶ molecules cm⁻³ at 10:00, Fig. 5.d) and the maximum $J_4$
      is one quarter of that calculated for Intense-NPF-CHC (2.17 cm⁻³ s⁻¹ at 11:00, Fig. 5.f).

From 12:00 to 17:00, the most notable difference to Intense-NPF-CHC is the influence of urban pollution from the city with
      eBC levels peaking at 0.65 µg m⁻³ at 12:00 (Fig. 5.c) and followed by larger Aitken particle concentrations ($N_{40-100}$;
      maximum of 5.82×10³ cm⁻³ at 13:00, Fig. 5.h), reflecting the advection of polluted urban emissions. The short-range air mass
      analysis (<4 h; Fig. 5.r, red spheres) suggests that the air masses are indeed coming from close to the surface and from the
      area of EAC at 12:00. As the afternoon progresses, a gradual decrease in pollution levels and particle concentrations is
observed, indicating the dispersion of pollutants. The concentration of $N_{13-40}$ peaks at 12.8×10³ cm⁻³ by 15:00 (Fig. 5.h) at the
      same time the eBC has plunged to 0.22 µg m⁻³ (Fig. 5.c). This suggests that the $N_{13-40}$ is not the result of primary emission
      advected from EAC, which would exhibit elevated concentrations of eBC, but rather of an ongoing regional NPF process that
      was briefly interrupted by the localized emission from EAC.

In the evening transition from 18:00 to 23:00 (at CHC), incident SW radiation diminishes to 4.70 W m⁻² by 18:00 (Fig. 5.a),
      and temperatures drop down to -0.4°C at 23:00 (Fig. S9), marking the establishment of the night-time residual layer. Particle
      concentrations across all size ranges start to decline, with $N_{100-440}$ reducing to 0.36×10³ cm⁻³ and $N_{13-40}$ to 4.25×10³ cm⁻³ by
      23:00 (Fig. 5.h). The decrease in eBC to 0.07 µg m⁻³ by 23:00 indicates a reduction in the advection of polluted air masses
      from the city (Fig. 5.c). Our short-range analysis of air mass origins (<4 h; red spheres in Fig. 5.s), shows these air masses
originate from the south, share a similar altitude with CHC, and are not affected by the pollution typically seen at EAC.



At *EAC*, comparing the diurnal patterns of the PNSD between Polluted-EAC and Intense-NPF-EAC reveals significant differences (Fig. 3.d and m). Notably, the growth observed in Intense-NPF-EAC is absent in Polluted-EAC. The daytime maximum concentration of $N_{4-13}$ ($20.3\times10^3$ cm$^{-3}$, Fig. 5.h) in Polluted-EAC is also reduced to one fifth as compared to
Intense-NPF-EAC. Furthermore, the decline in concentrations of larger particles $N_{40-100}$ and $N_{100-440}$ is delayed, occurring at 13:00 instead of 8:00 as observed in Intense-NPF-EAC which correlates well with eBC and indicates higher levels of pollution in the morning also at EAC. For example, at 12:00 the concentration of eBC is 0.22 µg m$^{-3}$ vs 0.94 µg m$^{-3}$ for Intense-NPF-EAC and Polluted-EAC, respectively.

Additionally, significant differences between Polluted-EAC and Intense-NPF-EAC are evident among the observed meteorological variables. For example, the minimum temperature before dawn is higher in Polluted-EAC compared to Intense-NPF-EAC (1.3°C vs. -0.6°C, respectively, Fig. 5.i and 4.i) as well as daily median WVMR (6.5 vs. 3.4 g kg$^{-1}$, Fig. S9). Moreover, in Polluted-EAC, the increase in wind speed starts later in the day, around 12:00, while in Intense-NPF-EAC, it becomes noticeable as early as 8:00 (see Fig. 5.n and 4.n). This difference likely explains why $N_{40-440}$ concentrations take
longer to decrease in Polluted-EAC than in Intense-NPF-EAC (previous paragraph). Analysing the short-term air mass history (<4 h), we observe that at 12:00, air masses in Polluted-EAC originate from areas closer to the surface, while in Intense-NPF-EAC the air masses appear to originate  from upper parts of the PBL, with source regions reaching altitudes comparable to CHC elevation (see 4.r and Fig. 5.r).

The observations consistently indicate weaker diurnal cycle in PBL height in Polluted-EAC, which contrasts with the conditions in Intense-NPF-EAC. Specifically, the PBL in Polluted-EAC grows at a slower rate and does not reach the same PBL height as in Intense-NPF-EAC. This delay is attributed to a higher proportion of insolation being used for latent heat processes, as evidenced by a higher WVMR, unlike the drier conditions in Intense-NPF. Furthermore, the pre-dawn measurements show that both the temperature is higher and the concentrations of $N_{40-440}$ and eBC are lower in Polluted-EAC
compared to Intense-NPF-EAC ($9.85\times10^3$ vs $17.4\times10^3$ cm$^{-3}$ for $N_{40-440}$, and 2.49 vs 4.67 µg m$^{-3}$for eBC, respectively). These differences suggest a deeper nocturnal layer in Polluted-EAC, which facilitates the dilution of pollutants more effectively than in Intense-NPF-CHC. In the latter scenario, higher pollutant concentrations likely result from their accumulation due to a shallower PBL. Additionally, increased cloudiness in Polluted-EAC (Fig. S6) contributes to both reduced nocturnal cooling and diminished daytime surface heating. This results in a less stable surface layer at night that is less prone to pollution
accumulation and a shallower PBL during the day that takes longer to dilute pollutants.

The formation rate $J_4$ in Polluted-EAC reaches its peak at 13:00 (9.2 cm$^{-3}$ s$^{-1}$, Fig. 5.m) and is only one fifth of the peak value estimated at Intense-NPF-EAC. At this time (13:00) the calculated CS is considerably reduced from $16.8\times10^3$ s$^{-1}$ (11:00) to $9.8\times10^3$ s$^{-1}$ (Fig. 5.l), which is comparable with the CS at Intense-NPF-EAC at the same time ($10.5\times10^3$ s$^{-1}$; Fig. 4.l).



However, the peak $J_4$ is still 4 times lower than at this same time on Intense-NPF-EAC (9.18 vs. 32.8 cm$^{-3}$ s$^{-1}$, Fig. 4.m and 5.m). Therefore, the difference between $J_4$ does not seem to be explained by higher losses and may indicate differences in the chemical environment between these two categories that favours increased nucleation and growth during Intense-NPF-EAC. The fact that incident SW radiation is similar (only measured at CHC but expected to be representative of both locations) at this time between these two categories  (878 vs 913 W m$^{-2}$ for Polluted-CHC and Intense-NPF-CHC, respectively) suggests

that different rates of photons and subsequent photooxidation do not explain the above-mentioned difference in $J_4$ (under the hypothesis of a similar chemical environment). The different provenance of the air masses in the short, medium and long-range between Polluted-EAC and Intense-NPF support the hypothesis of a differentiated chemical environment, with Polluted-EAC receiving air masses from the south-eastern edge of the Altiplano and Intense-NPF-EAC receiving air masses from the west centre Altiplano (Fig. 5.p-s and 8). The hypothesis can be further supported by the measured SA at CHC at

13:00 (potentially representative of EAC under the assumption the SO$_2$ is not locally emitted but transported), which is only one fifth in Polluted as compared to Intense-NPF (1.67×10$^6$ vs 9.02×10$^6$ molecules cm$^{-3}$, respectively).

### 1.7.2.1    Similarities and differences

The atmospheric conditions at EAC and CHC are different between EAC and CHC are different between the Polluted and

Intense-NPF category. especially during the daytime. In Intense-NPF, the predominant and synoptic westerly wind direction, combined with the south to north alignment of EAC and CHC, implies that air masses originating from the Altiplano simultaneously affect both stations with reduced direct air exchange between them (Fig. 8). However, in the Polluted category, the southerly short-range wind direction implies that the air masses reach the stations in sequence, first EAC and then the same air mass enriched by air pollution from EAC (or more specifically the La Paz-El Alto conurbation: orange

polygon in Fig. 1) reaches CHC shortly after. This is evident in the noon eBC peak observed at CHC that happens while a downward eBC trend is observed in EAC ( concentrations 0.94 and 0.65 µg m$^{-3}$ at 12:00, respectively). Furthermore, it should be mentioned that we not only observe a peak in eBC, but also peaks in particle phase organics, nitrate, and ammonium, all of which are indicators of anthropogenic pollution (1.4, 0.52, 0.57 µg m$^{-3}$, respectively; Fig. S11).

Despite the differences between the Intense-NPF and Polluted, in Polluted we still observe nucleation at both locations (calculated max $J_4$ is 9.2 and 2.2 cm$^{-3}$ s$^{-1}$ for CHC and EAC, respectively). Furthermore, it could be argued that at CHC, if we discard the hours when intense city pollution is observed, we observe a regional NPF event with nucleation and growth in the morning and the result growing mode upwind observed late in the afternoon. Based on this assumption, the final size of the particles is smaller than that observed in the case of Intense-NPF-CHC. For example, at 17:00 the centre of the observed

growing mode is 31.6 nm at Intense-NPF-CHC and only 22.4 nm at Polluted-CHC. This is likely the result of a different chemical environment with different source regions (Fig. 8). For example, the maximum concentration of SA, which happens at 10:00 in both scenarios, is lower during Polluted-CHC compared to Intense-NPF-CHC (6.05×10$^6$ and 12.8×10$^6$ molecules cm$^{-3}$, respectively)

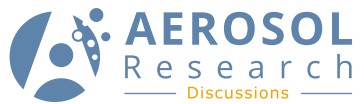

### 1.7.3 Volcanic category

The Volcanic category consists of a total of 9 days. During these days, at both locations the FLEXPART air mass analysis shows that the air masses come from the northern Altiplano, but intrusions from lowland regions north of CHC and EAC are also common, either in the medium range through the Sorata Valley, or in the medium range via the valley to the north in the eastern ridge of the Altiplano (Fig. 8). Even though air masses traverse the Altiplano on their way to CHC and EAC, the high values of RH recorded during these days when compared to Intense-NPF further hints to the stronger influence of humid air

masses from the lowlands (75% vs 45% for Volcanic- and Intense-NPF-EAC and 54% vs 34% for Volcanic- and Intense-NPF-CHC, respectively; Fig. S9). The most outstanding feature during these days is the observed high concentration of sulphate (and aerosol total volume) both at EAC and CHC. At EAC, it is generally high with a daily median of 1.34 $\mu$g m$^{-3}$, and particularly high between 10:00 and 18:00 (1.80 $\mu$g m$^{-3}$). At CHC a constant high value is observed through the day with a median of (1.94 $\mu$g m$^{-3}$; Fig. 6 and Table 1). Furthermore, a comparatively high and constant median value of CS is present

before 10:00 (i.e. before the growing particles influence the CS) at CHC ($7.7\times10^{-3}$ vs $3.3\times10^{-3}$ s$^{-1}$ for Volcanic-CHC and Intense-NPF-CHC, respectively). Finally, the highest peak of SA is observed in this category at CHC ($35.9\times10^6$ molecules cm$^{-3}$ at 11:00) although SA measurements were available on only 3 of the 9 days.

Nucleation which occurred at both CHC and EAC stations was identified on 8 and 9 days (respectively) in our traditional

classification. Subsequent growth was observed in 6 and 7 days at EAC and CHC respectively (Fig. S4). At CHC, the high concentration of SA does not translate into a correspondingly intense J4. For instance, at Volcanic-CHC, the median noon values for SA and J4 and are $24.4\times10^6$ molecules cm$^{-3}$ and 0.7 cm$^{-3}$ s$^{-1}$ respectively. In contrast, at Intense-NPF-CHC, these values are $9.47\times10^6$ molecules cm$^{-3}$ for SA and 6.2 cm$^{-3}$ s$^{-1}$ for $J_4$. The smaller $J_4$ from higher SA concentration on Volcanic-CHC is qualitatively in agreement with higher CS ($10.0\times10^{-3}$ vs $7.6\times10^{-3}$ s$^{-1}$). The discrepancy between Volcanic and

Intense-NPF remains also when only the days with available SA data are compared (noon medians for 3 Volcanic days: SA = 24.4 molecules cm$^{-3}$, $J_4$ = 0.4 cm$^{-3}$ s$^{-1}$, CS = $11.4\times10^{-3}$ s$^{-1}$; and for 10 Intense-NPF days: SA = 9.47 molecules cm$^{-3}$, $J_4$ = 5.5 cm$^{-3}$ s$^{-1}$, CS = $6.7\times10^{-3}$ s$^{-1}$). Whether other factors are needed to explain the drastic difference in $J_4$ remains unexplained. One hypothesis is that the absence of other required nucleating bases (e.g., NH$_3$; Zha et al., 2023) may contribute to the lower $J_4$ values. We have found that during nucleation, at CHC, the ratio between $N_{2-4}^-$ and $N_{2-4}^+$ is reduced to 0.02 (it is 0.37 the rest

of the days; Fig. S17). This change in ratio is not observed at EAC and remains constant at ~1.14. Higher RH may also play a role but the effects of RH in formation rates are varied across different studies (Kerminen et al., 2018).

At *CHC*, the diurnal pattern reveals notable similarities in particle formation between Volcanic-CHC and Intense-NPF-CHC during the predawn and early morning hours (00:00-11:00). According to the PNSD, both locations exhibit a similar

nucleation process and early growth phase. This process starts at approximately 9:00 with a growth rate of GR$_{4-7}$ about 6.5 nm h$^{-1}$ (Table S1). However, the process is much less intense in Volcanic-CHC. This less intense NPF process is also





reflected in a smaller $J_4$ (as noted above) resulting in a reduction of the maximum concentration $N_{7-13}$ and $N_{13-40}$ ($11.9\times10^3$ and $30.5\times10^3$ cm$^{-3}$ at Volcanic- and Intense-NPF-CHC, respectively). The incident SW radiation pattern is also similar during the early morning which depicts clear sky conditions, however after 9:00 small perturbations are observed at

Volcanic-CHC indicating the presence of clouds (Fig. 6.a) and furthermore the observed CF at 14:00 is 59% (Fig. S6). Another important difference to Intense-NPF is found in $N_{100-440}$. This value remains relatively constant during the day. At Volcanic-CHC, it is twice as high as at Intense-NPF-CHC, with daily median values of $1.38\times10^3$ and $0.63\times10^3$ cm$^{-3}$, respectively. The higher concentration of $N_{100-440}$ particles, and notably in the overall particle mass, likely stems from primary emissions of Sabancaya Volcano or from particles formed secondarily through the reaction of SO$_2$ emitted by the

volcano during its transport to CHC. Our air mass origin analysis reveals that the SRR contribution from a 50km radius zone cantered around Sabancaya exceeds others by over 5 times (SRR= 0.5 h, not shown). The increased CF during these days may also contribute to in-cloud aqueous-phase conversion of SO$_2$ to sulphate. Regarding the local meteorology, the daytime wind patterns are similar to Intense-NPF-CHC both in term of speed and direction (Fig. 4.g and Fig. 6.g).

At *EAC*, the diurnal patterns of Volcanic-EAC and Intense-NPF-EAC (Fig. 3.m and q) show consistently similar characteristics. Specifically, the maximum diurnal concentration of small particles $N_{4-7}$ and $N_{7-13}$ exhibit remarkable similarity between the two categories ($42.3\times10^3$ and $35.3\times10^3$ cm$^{-3}$ at Volcanic-EAC, and $48.8\times10^3$ and $39.9\times10^3$ cm$^{-3}$ at Intense-NPF-EAC, respectively). This similarity in the smallest sizes is also reflected in comparable $J_4$ (52.1 vs. 50.7 cm$^{-3}$ s$^{-1}$ for Volcanic-EAC and Intense-NPF-EAC, respectively). However, at the intermediate sizes $N_{13-40}$, the number concentration

in Volcanic-EAC is similar to Intense-NPF-EAC only until noon. Afterwards it is significantly lower, particularly during the early afternoon (14:00), when the concentration is halved ($13.4\times10^3$ vs $30.9\times10^3$ cm$^{-3}$). Furthermore, in the larger sizes $N_{100-440}$, the pattern is reversed, with the concentration at Volcanic-EAC slightly exceeding that observed at Intense-NPF-EAC ($1.38\times10^3$ vs $1.12\times10^3$ cm$^{-3}$, respectively). This is consistent with an almost tripled mass concentration of sulphates in Volcanic-EAC vs. Intense-NPF-EAC (1.93 vs 0.77 μg m$^{-3}$ at 14:00).


Meteorologically, both Volcanic-EAC and Intense-NPF-EAC experience similar wind speeds throughout the day, diverging in wind direction before 9:00; Volcanic-EAC shows a southwestern flow, while Intense-NPF-EAC has a north-western one. This early morning variation also impacts RH, with Volcanic-EAC exhibiting higher levels than Intense-NPF-EAC (92% vs. 69% before sunrise). These differences are attributed to Intense-NPF-EAC receiving a drier air mass from the northern

mountains (near CHC), while Volcanic-EAC being influenced by a humid air mass from the La Paz canyon. The moist conditions at Volcanic-EAC likely slow the PBL morning development due to higher surface latent heat fluxes, delaying pollutant dilution. This is also reflected in the CS values measured at 10:00 —$21.7\times10^{-3}$ s$^{-1}$ for Volcanic-EAC and $15.9\times10^{-3}$ s$^{-1}$ for Intense-NPF-EAC.



### 1.7.3.1 Similarities and differences

The comparison between CHC and EAC within Volcanic shows similarities to Intense-NPF, with a distinct NPF process observed at CHC and pronounced nucleation and growth at EAC in both categories. However, a key difference is the $J_4$ ratio between CHC and EAC: 1 to 5 in Intense-NPF and 1 to 50 in Volcanic. This suggests that in this environment volcanic conditions tend to reduce NPF intensity – as compared to Intense-NPF – more at CHC than at EAC. Before noon, the smaller-sized particles follow a similar pattern to that of $J_4$ with e.g. maximum concentration of $N_{4-7}$ reaching $42.3 \times 10^3$ cm$^{-3}$

vs. $1.19 \times 10^3$ cm$^{-3}$ at EAC, and CHC, respectively. These differences are significantly higher than in Intense-NPF, implying that urban NPF is more affected by local emissions in Volcanic. Together, the results indicate that volcanic plumes do not result in as strong NPF events despite the prevalence of high SA possibly by higher CS (in both CHC and EAC) and this effect is exacerbated in CHC likely due to limited availability of relevant gaseous precursors, resulting from anthropogenic activities.


Afternoon particle measurements reveal a different pattern for the Aitken mode ($N_{13-100}$), where the concentrations at EAC and CHC become similar in magnitude. For example, at 15:00, $N_{13-40}$ concentrations are $7.11 \times 10^3$ cm$^{-3}$ at CHC and $11.5 \times 10^3$ cm$^{-3}$ at EAC, and $N_{40-100}$ concentrations are $2.58 \times 10^3$ cm$^{-3}$ at CHC and $3.10 \times 10^3$ cm$^{-3}$ at EAC. This similarity in magnitude between CHC and EAC in the afternoon is also observed in Intense-NPF. A likely explanation for this shift is that morning

conditions favour local NPF at EAC (higher nucleation mode, $N_{7-13}$, concentrations), but as the day evolves and the PBL deepens, both stations likely share the same air mass (Fig. 6.r). Consequently, the similarities observed between the stations in afternoon particle number concentrations, specifically in the growing NPF mode size range, suggest they are the result of regional NPF events within the well mixed and deep PBL.

### 1.7.4 Cloudy category

Finally, we briefly discuss the last category denoted *Cloudy*. A total of 22 days belong to this category, which is characterized by a predominance of overcast days (Fig. 7.a) and frequent cloud coverage in the domain (Fig. S6). At both locations, the FLEXPART air mass history analysis (Fig. 7.p) shows that air masses originate in the lowlands situated to the east of the stations, encompassing both the northern (Western Amazonia) and southern (Chaco) regions. This group stands out from the others by its elevated relative humidity (median values of 75% and 73% for EAC and CHC, respectively).

Nucleation is observed on most days at EAC (20) but only in 7 days at CHC. Growth is observed in 2 days at both stations (Fig. S4).

At *CHC*, the PNSD (see Fig. 1.g) shows diminished particle concentrations across all sizes throughout the day compared to the other categories . Similarly, the SA concentration is also lowest of the 4 categories (diurnal maximum is $3.44 \times 10^6$

molecules cm$^{-3}$), likely due to lower photochemical activity and/or possibly lower SO$_2$ concentrations (not measured).



At *EAC* the observed diurnal pattern closely resembles Polluted-EAC both in the meteorology as well as in the PNSD. Furthermore, a comparable $J_4$ is observed (6.1 cm$^{-3}$ s$^{-1}$) . The primary difference between Cloudy-EAC and Polluted-EAC is the earlier increase in wind speed observed at Cloudy-EAC. For instance, by 12:00, the wind speed at Cloudy-EAC reaches

3.7 m s$^{-1}$, while at Polluted-EAC it is only 2.5 m s$^{-1}$. This translates into an early decrease in CS at Cloudy-EAC beginning at 9:00 (Fig. 7.l). At Polluted-EAC, this decrease does not start until after 12:00 (Fig. 5.l).

In this category, we notice a distinct pattern not seen in the others: during the afternoon, Aitken mode particle concentrations are not similar at the two stations. Instead, concentrations are significantly higher at EAC compared to CHC. Specifically,

for $N_{13\text{-}40}$ and $N_{40\text{-}100}$, the concentrations at EAC are 5.3×10$^3$ cm$^{-3}$ and 1.6×10$^3$ cm$^{-3}$, respectively, while at CHC, they are only 0.6×10$^3$ cm$^{-3}$ in both size ranges. This discrepancy suggests two possibilities: either the mixing layer does not extend as high as CHC, isolating it more from surface influence, or the air masses have different sources. The air mass analysis supports both ideas, indicating minimal surface influence in this category (not shown) and air masses arriving at CHC via the northwest ridge and at EAC via the La Paz valley southeast of the city (Fig. 8). Furthermore, both EAC and CHC share the

same long-range origin for their air masses (>200 km; Fig. 8). However, nucleation events are seen at EAC and not at CHC suggesting that EAC likely has a sufficient concentration of precursor gases from local emissions to support this level of nucleation activity, even in the absence of a regional NPF event.

## 5    Conclusions

Our study examines NPF in the Andes, specifically around the eastern edge of the Bolivian central Altiplano, likely

representative of similar Andean regions. This is the first regional analysis to simultaneously compare NPF events at two interconnected sites with different altitudes: EAC, an urban site on the eastern edge of the Altiplano, and CHC, a mountain site in the Andean Cordillera Real range (19 km north of EAC). This dual-site approach allowed us to explore several aspects of NPF in the Andean highlands, including how NPF characteristics vary with altitude, the impact of volcanic degassing, and anthropogenic influences from nearby urban areas. We combined the recently developed "nanoparticle

ranking analysis" (Aliaga et al., 2023) with k-means clustering to categorize days based on the intensity of NPF (determined as the daily maximum $N_{4\text{-}7}$) simultaneously observed at CHC and EAC. We obtained four significantly different categories of days:

- *Intense-NPF* (18 days; $N_{4-7}^{CHC}$=11.5×10$^3$ cm$^{-3}$; $N_{4-7}^{EAC}$= 62.1×10$^3$ cm$^{-3}$),
- *Polluted* (16 days; $N_{4-7}^{CHC}$=4.9×10$^3$ cm$^{-3}$; $N_{4-7}^{EAC}$= 14.8×10$^3$ cm$^{-3}$),

- *Volcanic* (9 days; $N_{4-7}^{CHC}$=1.5×10$^3$ cm$^{-3}$; $N_{4-7}^{EAC}$= 62.3×10$^3$ cm$^{-3}$), and
- *Cloudy* (22 days; $N_{4-7}^{CHC}$=0.5×10$^3$ cm$^{-3}$; $N_{4-7}^{EAC}$= 11.2×10$^3$ cm$^{-3}$),



named based on their emergent characteristics at both sites. These four categories allowed us to conduct a detailed exploration of the impact of meteorology, SA (only CHC), and volcanic degassing on the formation of new particles in the region, as well as to compare the NPF processes observed in EAC and CHC.


We have found that the intensity of NPF events varies depending on the geographical location, levels of anthropogenic influence and/or elevation with EAC experiencing stronger events compared to CHC (about on order of magnitude but enhanced during Volcanic and reduced during Polluted). This indicates that the characteristics of NPF in the region are likely influenced by local anthropogenic activities and/or surface influences. The two stations we considered vary in altitude and

local pollution levels, complicating our understanding of how NPF intensity would appear in a station differing in only one of these aspects. For instance, it remains uncertain what the NPF intensity would be at a station with the same altitude as EAC but without high levels of precursor emissions from anthropogenic activities (background Altiplano). We hypothesize that the intensity at such a station would likely fall between those observed at EAC and CHC.

We have observed that NPF events typically begin around the same time in both CHC and EAC (9:00), indicating a consistent pattern of event initiation. While the specific triggers for NPF event onset are not yet fully understood, our findings suggest that multiple factors may play a role. At CHC, factors such as increased solar radiation and surface influence appear to influence the onset of NPF events. Conversely, at EAC, factors such as increased solar radiation, changes in wind direction (from easterlies to westerlies), decrease in RH and the intrusion of air masses from above the stable

nocturnal PBL contribute to event initiation. Almost all these factors are correlated and modulated by solar activity.

We observed that intense NPF events (Intense-NPF) start in both a rapidly growing PBL (EAC) and a residual layer with increasing influence from the surface (CHC) with both sites affected by air masses coming from the Altiplano. These events gradually merge throughout the day, forming a regional mixture across the PBL. This indicates that the formation of NPF

events is not confined to specific atmospheric layers but rather involves complex interactions across different layers of the atmosphere. We do not observe any event in the free troposphere as during event days and in most of the time except for brief periods at dawn, our air mass analysis suggests that CHC is heavily influence by the residual layer.

At *EAC*, daytime increase of nucleation mode particle concentrations is observed every day regardless of airmass origin.

This increase during daytime is likely not the result of primary emissions from anthropogenic emissions as they are anticorrelated with other markers of anthropogenic pollution such as eBC. It is likely that the combined effect of precursors emitted from anthropogenic activities and/or (less likely) precursors emitted from the surface which rapidly photo oxidate and nucleate with increased radiation and rapidly increase in concentration. The intensity of the NPF process does however vary with airmass origin. Formation rates are considerably lower when air masses arrives from the La Paz valley ($J_4^{\text{Polluted}}=$





9.2 cm$^{-3}$ s$^{-1}$; $J_4^{Cloudy}$= 6.1 cm$^{-3}$ s$^{-1}$) and are increased when the air masses comes from the Altiplano ($J_4^{Intense-NPF}$= 50.7 cm$^{-3}$; $J_4^{Volcanic}$= 52.1 cm$^{-3}$ s$^{-1}$). Increased volcanic influence does not modify the observed formation rate. Finally, on 41% percent of the observed days (Intense-NPF and Volcanic), the daytime population of $N_{4-13}$ particles reached approximately 1×10$^5$ cm$^{-3}$. This poses potential health risks (Pedata et al., 2015) to the densely populated conurbation of El Alto/La Paz.

At *CHC*, a new nucleation mode is observed often but not always (74% of observed days) and is sensitive to airmass origin with fewer events observed when the air masses come from the lowlands (Cloudy). The intensity of NPF is also sensitive to airmass origins with higher formation rates observed when air masses are transported from the Altiplano ($J_4^{Intense-NPF}$= 7.7 cm$^{-3}$ s$^{-1}$). However, when these air masses are also influenced by the volcanic activity, formation rates are drastically reduced ($J_4^{Volcanic}$= 1.2 cm$^{-3}$ s$^{-1}$). The absence of a stabilizing base cluster may play a crucial role in this aspect as during these days
the concentration $N_{2-4}^+$ is drastically reduced ($N_{2-4}^{+Volcanic}$=19 cm$^{-3}$; $N_{2-4}^{+Intense-NPF}$=123 cm$^{-3}$).

The anthropogenic influence from the city plays a role in interrupting observed events at CHC (Polluted). During these days, an ongoing event is interrupted at noon ($J_4^{11:00}$=2.2 cm$^{-3}$ s$^{-1}$; $J_4^{13:00}$=0.9 cm$^{-3}$ s$^{-1}$). Between 12:00 and 15:00, similar values of eBC, and slightly higher $N_{40-440}$ are observed at CHC compared to EAC. A shallower PBL than Intense-NPF and transport
dominated by thermal driven winds prevails during these days. The fact that the arrival of this plume, with potentially less entrainment from the layer above the PBL and which reduces the intensity of NPF seems to suggest that the intensity of NPF events benefits from entrainment (e.g. SA is reduced from 6.1×10$^3$ molecules cm$^{-3}$ at 10:00 to 1.7×10$^3$ molecules cm$^{-3}$ at 13:00). Increased CS does not seem to explain the sudden reduction in formation rates as it is comparable in magnitude to the same time during Intense-NPF when the formation rates are 7 times higher.

The ongoing NPF process does not result in an increase in accumulation mode particles (100-440 nm) across any category or site. At CHC, increases are only observed during Polluted at 12:00, likely from advection of anthropogenic-influenced air masses from the city. The necessary growth rate would need to be around 25 nm h$^{-1}$, well beyond the observed ~7 nm h$^{-1}$. Typically, any increase in particles >40 nm before early afternoon is likely due to primary emissions or changing air masses,
not NPF.

However, in the late afternoon, we do see a significant increase in larger Aitken mode particles (40-100 nm) during Intense-NPF days (which accounts for 36% of the total measured days), suggesting a link to the ongoing NPF process, coinciding with the observed growth rate. At EAC, it increases from 3.2×10$^3$ cm$^{-3}$ at 4:00 to 4.9×10$^3$ cm$^{-3}$ at 17:00, although this
increase may be overestimated due to primary emissions from the city. At CHC, the count rises six-fold from 1.1×10$^3$ cm$^{-3}$ at 8:00 to 6.2×10$^3$ cm$^{-3}$ at 16:00. Given that these particles form between 4 and 5.5 km and likely drift over the eastern lowlands, they could impact the region's climate by modifying cloud properties.



# Acknowledgements

–     DA wishes to thank the scientists, technicians and personnel involved at the Chacaltaya station, whose outstanding commitment enables high-quality atmospheric observations in a challenging environment.

–     The personnel of the Instituto de Investigaciones Físicas of Universidad Mayor de San Andrés.

–     Doctoral School in Atmospheric Sciences at the University of Helsinki (ATM-DP).

–     ChatGPT (GPT-3.5;4, OpenAI's large-scale language-generation model) has been used to improve the writing style of some excerpts in this article. DA reviewed, edited, and revised the ChatGPT generated texts to his own liking

and takes ultimate responsibility for the content of this publication.

–     DA thanks Sara Blichner for valuable discussions, input, and support during the writing process.

# Financial support

–     ACCC Flagship funded by the Academy of Finland (grant nos. 337549 (UH) and 337552 (FMI))

–     Academy Professorship funded by the Academy of Finland (grant no. 302958)

–     Academy of Finland (grant nos. 1325656, 311932, 334792, 316114, 325647, 325681, 347782, and 337549)

–     "Quantifying carbon sink, CarbonSink+ and their interaction with air quality", INAR project funded by the Jane and Aatos Erkko Foundation

–     "Gigacity" project funded by the Wihuri Foundation

–     INAR project funded by Jane and Aatos Erkko Foundation

–     European Research Council (ERC) project ATM-GTP (grant no. 742206)

–     European Commission, H2020 Research Infrastructures (CHAPAs grant no. 850614; FORCeS grant no. 821205)

–     Horizon Europe Infrastructures (FOCI grant no.101056783).

–     Knut and Alice Wallenberg (KAW) foundation

–     FAPESP – Fundação de Amparo à Pesquisa do Estado de São Paulo through grants 2017/17047-0 and 2023/04358-

950      9.

–     Continuous observations at CHC and EAC are supported by the Institute de Recherche pour le Développement (IRD) France and IRD delegation in Bolivia, by Observatoire de Sciences de l'Univers de Grenoble (OSUG) partly through Labex OSUG@2020 (ANR10780 LABX56), and by CNRS/INSU and Ministère de l'Enseignement Supérieur et de la Recherche with contribution to ACTRIS-FR.

–     University of Helsinki support via ACTRIS-HY.



## Author contributions

- DA prepared and edited the manuscript, with significant input from VAS and RK, and contributions from MA, PA, LB, RC, SC, YG, LH, DHR, WH, VMK, AMK, MK, PL, VMB, CM, IM, PP, WS, KS, LT, GU, FV, CW, CX, QZ and FB.

- DA performed the analysis.
- LH and SC performed the ACSM measurements during SALTENA and ACSM data preparation.
- MA, PL, AW, RK, KS, LB contributed to the development of the GAW station CHC.
- FB, MK, CM, RK, PA, AW, MA, PL, KS, DA, and QZ contributed to the overall planning of the SALTENA campaign.

- DA, SC, YG, LH, WH, AMK, VMB, CM, IM, WS, GU, FV, CW, CX, QA, and FB performed the measurements during the SALTENA campaign.
- PP, DW, RC provided valuable and significant input during the data analysis.
- FB, VAS and RK conceived the study and led the overall scientific investigation.

**Competing interests**

Markku Kulmala is a member of the editorial board of journal Aerosol Research.



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



# 6 Figures

l260 **1.8**

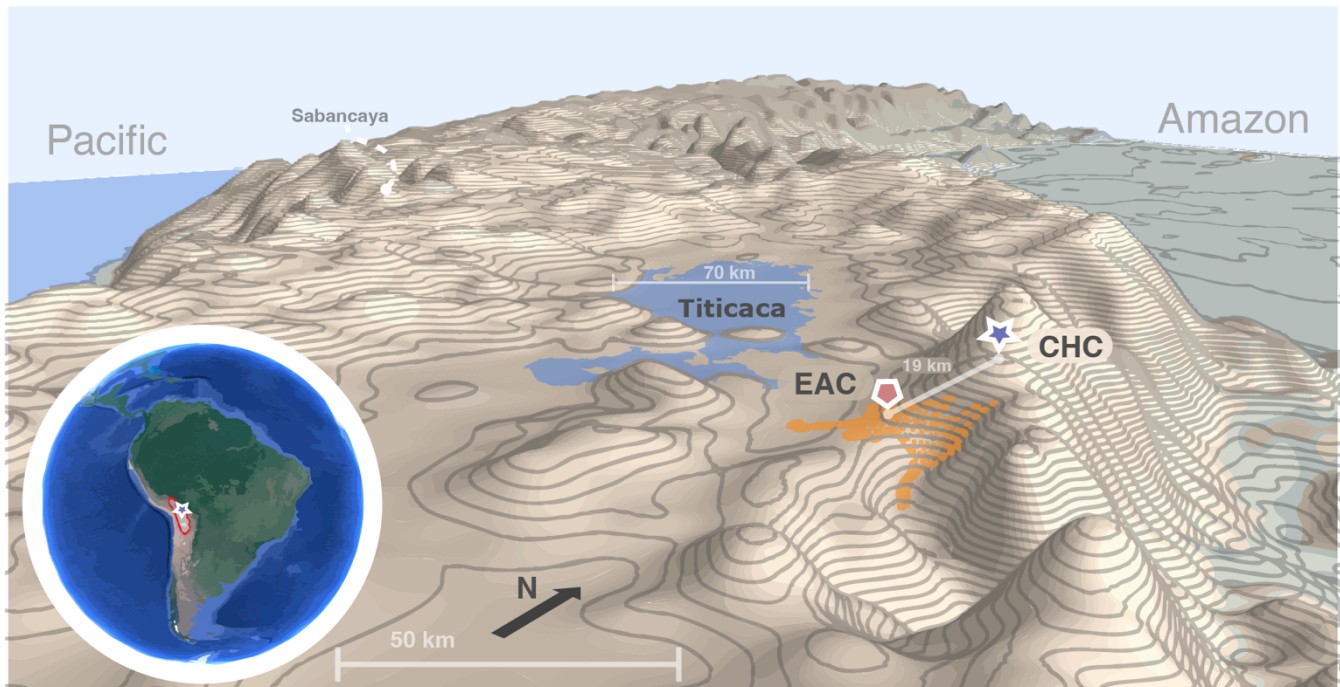

**Figure 1: The illustration shows a general description of the region showing including a world map (lower left) and a topographic portrayal of the area. On this world map, CHC is denoted with a star, and a red line outlines the Altiplano plateau. On this topographic portrayal, The Chacaltaya station (CHC) is marked with a lilac star, while the El Alto City (EAC) station is marked**
l265 **with a pink pentagon. The conurbation of El Alto-La Paz area is coloured in orange. Additionally, distinct features of the region are displayed, including: the La Paz valley situated southeast of the EAC, Lake Titicaca to the west of the stations, the Sabancaya volcano to the west of the stations, and lastly, the Amazon region and the lowlands to the east and the Pacific Ocean to the west.**



**1.9**

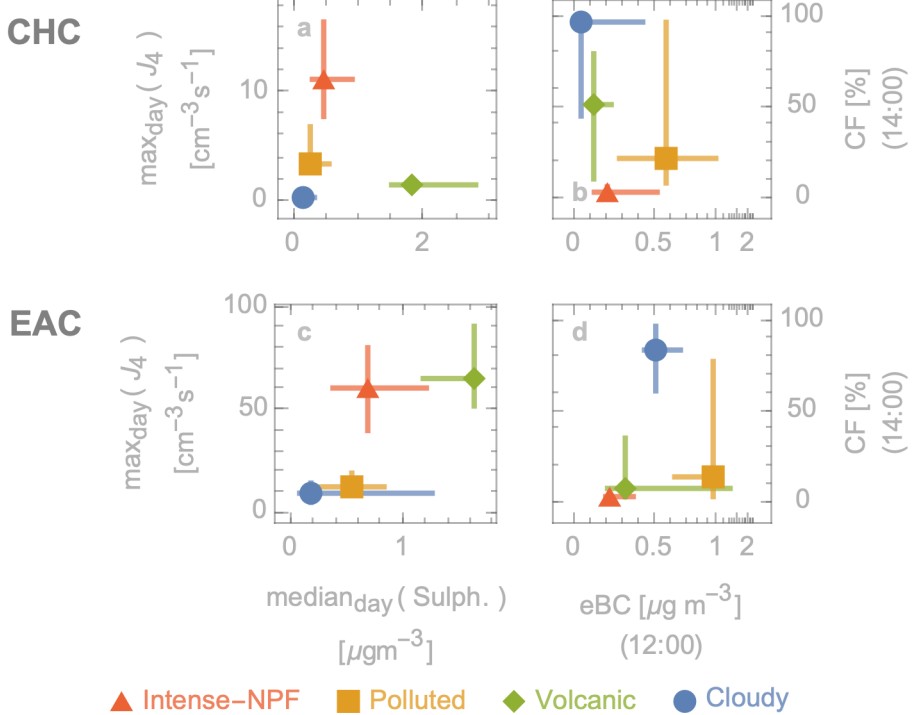

**Figure 2: Key parameters for naming the categories derived using k-means clustering on $N_{4-7}^{max}$ at CHC and EAC. The top panels (a, b) display variables observed at CHC, while the bottom panels (c, d) present EAC observations. Panels a and c show daily maximum $J_4$ values (y-axis) against median daily sulphate concentrations (x-axis). Panels b and d illustrate cloud fraction (CF) percentages at 14:00 within a 10 km radius of each station (y-axis), as captured by MODIS, versus eBC concentrations at 12:00 (x-axis), when peak eBC levels are observed at CHC and comparatively (to other categories) high levels are seen at EAC. The markers indicate median values, and the lines represent the interquartile range (IQR).**




1.10



**Figure 3: The central panel i displays the daily maximum values of particle concentration in the range of 4 to 7 nm for EAC (x-axis) and CHC (y-axis) (see methods). These points are grouped into four categories or quadrants using k-means clustering. Each group of days is named after its key emergent parameter (Fig. 2): Intense-NPF (upper right; j-m), Polluted (upper left; a-d), Volcanic (lower right; n-q), and Cloudy (lower left; e-h). The panels in the four corners show the daily median values of particle concentration N4-7 (first row) and PNSD (second row) for CHC (first column) and EAC (second column). Panel (r) shows the group that is assigned to each day (grey is used for days with insufficient data) and whether nucleation (dot; Nuc-D) and/or growth (circle; Gr-D) was observed at the stations.**





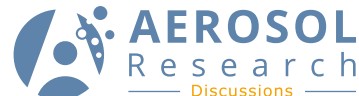

**1.11**

Figure 4: **Main characteristics of the Intense-NPF category. The top left and right panels provide information on diurnal median values of key variables at CHC and EAC, respectively. The shadings show the inter-quantile range and the grey line is the median of all categories (shown for comparison). Panels g and n show the wind speed and the arrows point to the median direction (up means wind coming from south to north and right, wind coming from west to east). Panels h and o show the concentration at different ranges color-coded and specified in the legend above. Panel p shows the median long-range air mass origins for CHC (Source-Receptor relationship; SRR). The sky-blue half disk represents the location of CHC (with EAC in orange close to the same**



**location). The lilac circle marks the location of Volcano Sabancaya. The EAC long-range air mass origin is very similar and is presented in Figure 8. Panels q-s, depict the short-range (<4h) air mass origins for a representative day at three key hours: 7, 12,**

**and 21 (see Methods for calculation). The red spheres indicate the location of the air mass arriving at CHC, with their size representing the quantity and/or duration of air tracers in that location. The grey spheres provide the same information, but for EAC. The green arrow points to north. CHC is denoted by a pyramid with a red stick on top while EAC is marked with a cube topped by a grey stick. Each grid line is separated by approximated 25 km.**



**1.12**

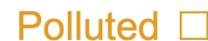


**Figure 5: Main characteristics of the Polluted category. The top left and right panels provide information on diurnal median values of key variables at CHC and EAC, respectively. The shadings show the inter-quantile range and the grey line is the median of all categories (shown for comparison). Upper right grey integers mark the number of available days for the given variable. Panels g and n show the wind speed and the arrows point to the median direction (up means wind coming from south to north and** 1310 **right, wind coming from west to east). Panels h and o show the concentration at different ranges color-coded and specified in the legend above. Panel p shows the median long-range air mass origins for CHC (Source-Receptor relationship; SRR). The sky-blue**





**half disk represents the location of CHC (with EAC in orange close to the same location). The lilac circle marks the location of Volcano Sabancaya. The EAC long-range air mass origin is very similar and is presented in Figure 8. Panels q-s, depict the short-range (<4h) air mass origins for a representative day at three key hours: 7, 12, and 21 (see Methods for calculation). The red spheres indicate the location of the air mass arriving at CHC, with their size representing the quantity and/or duration of air tracers in that location. The grey spheres provide the same information, but for EAC. The green arrow points to north. CHC is denoted by a pyramid with a red stick on top while EAC is marked with a cube topped by a grey stick. Each grid line is separated by approximated 25 km.**






l320   **1.13**



**Figure 6: Main characteristics of the Volcanic category. The top left and right panels provide information on diurnal median values of key variables at CHC and EAC, respectively. The shadings show the inter-quantile range and the grey line is the median of all categories (shown for comparison). Upper right grey integers mark the number of available days for the given variable.**

l325   **Panels g and n show the wind speed and the arrows point to the median direction (up means wind coming from south to north and right, wind coming from west to east). Panels h and o show the concentration at different ranges color-coded and specified in the legend above. Panel p shows the median long-range air mass origins for CHC (Source-Receptor relationship; SRR). The sky-blue**




**half disk represents the location of CHC (with EAC in orange close to the same location). The lilac circle marks the location of Volcano Sabancaya. The EAC long-range air mass origin is very similar and is presented in Figure 8. Panels q-s, depict the short-range (<4h) air mass origins for a representative day at three key hours: 7, 12, and 21 (see Methods for calculation). The red spheres indicate the location of the air mass arriving at CHC, with their size representing the quantity and/or duration of air tracers in that location. The grey spheres provide the same information, but for EAC. The green arrow points to north. CHC is denoted by a pyramid with a red stick on top while EAC is marked with a cube topped by a grey stick. Each grid line is separated by approximated 25 km.**



**1.14**



**Figure 7: Main characteristics of the Cloudy category. The top left and right panels provide information on diurnal median values of key variables at CHC and EAC, respectively. The shadings show the inter-quantile range and the grey line is the median of all categories (shown for comparison). Upper right grey integers mark the number of available days for the given variable. Panels g and n show the wind speed and the arrows point to the median direction (up means wind coming from south to north and right, wind coming from west to east). Panels h and o show the concentration at different ranges color-coded and specified in the legend above. Panel p shows the median long-range air mass origins for CHC (Source-Receptor relationship; SRR). The sky-blue half**





**disk represents the location of CHC (with EAC in orange close to the same location). The lilac circle marks the location of Volcano Sabancaya. The EAC long-range air mass origin is very similar and is presented in Figure 8. Panels q-s, depict the short-range**

**(<4h) air mass origins for a representative day at three key hours: 7, 12, and 21 (see Methods for calculation). The red spheres indicate the location of the air mass arriving at CHC, with their size representing the quantity and/or duration of air tracers in that location. The grey spheres provide the same information, but for EAC. The green arrow points to north. CHC is denoted by a pyramid with a red stick on top while EAC is marked with a cube topped by a grey stick. Each grid line is separated by approximated 25 km.**






## 1.15



**Figure 8. Illustrates the median of the origins of the air masses directed towards CHC and EAC for each category. The top two row provide an overview of the long-range air mass patterns associated with CHC and EAC. Rows three and four present a detailed view of the air mass patterns specific to CHC and EAC, respectively. CHC and EAC are shown by a sky blue and orange (half) disk, respectively. The Sabancaya volcano is marked by a lilac circle. The colour coded cells show the magnitude of source receptor relationship (SRR=$n \times r$) at that cell which quantifies the number ($n$) of passive air traces at the cell multiplied by their residence time ($r$) integrated over the previous 4 days. A cell with a value of 1 h would mean that all the air tracers spent the last 4 hours at that cell.**





# 7 Tables

## 1.16

| | Intense-NPF | | Polluted | | Volcanic | | Cloudy | |
|---|---|---|---|---|---|---|---|---|
| | CHC | EAC | CHC | EAC | CHC | EAC | CHC | EAC |
| $J_4^{max\ day}$ [cm$^{-3}$s$^{-1}$] | $7.7_{3.8}^{12.6}$ | $50.7_{35.4}^{71.8}$ | $2.2_{1.0}^{3.5}$ | $9.2_{5.4}^{13.6}$ | $1.2_{0.9}^{1.6}$ | $52.1_{33.3}^{90.9}$ | $0.2_{0.2}^{0.3}$ | $6.1_{3.5}^{11.5}$ |
| $SA^{max\ day}$ [10$^6$mol.cm$^{-3}$] | $12.8_{7.1}^{23.2}$ | | $6.1_{4.8}^{6.8}$ | | $35.9_{30.1}^{41.6}$ | | $3.4_{2.3}^{7.6}$ | |
| $GR_{4-7}$ [nm h$^{-1}$] | $7.0_{4.3}^{10.1}$ | $9.0_{6.5}^{10.3}$ | $6.8_{4.8}^{9.5}$ | $7.0_{5.9}^{11.1}$ | $6.2_{5.2}^{18.2}$ | $7.3_{6.8}^{8.4}$ | $7.7_{6.1}^{16.7}$ | $5.1_{2.7}^{12.0}$ |
| $CS^{10:00}$ [10$^{-3}$s$^{-1}$] | $4.0_{3.2}^{6.0}$ | $11.9_{10.5}^{15.9}$ | $3.8_{2.8}^{4.1}$ | $14.5_{13.6}^{19.9}$ | $7.5_{6.6}^{7.8}$ | $14.9_{13.8}^{16.5}$ | $1.3_{1.1}^{3.4}$ | $9.8_{7.9}^{15.3}$ |
| $N_{4-7}^{max\ day}$ [10$^3$cm$^{-3}$] | $11.5_{9.7}^{14.3}$ | $62.1_{51.8}^{75.6}$ | $4.9_{2.9}^{7.5}$ | $14.8_{10.9}^{18.2}$ | $1.5_{1.1}^{1.7}$ | $62.3_{49.6}^{70.6}$ | $0.5_{0.4}^{0.7}$ | $11.2_{5.9}^{15.2}$ |
| $N_{7-13}^{03:00}$ [10$^3$cm$^{-3}$] | $0.3_{0.1}^{0.5}$ | $1.8_{1.3}^{2.1}$ | $0.3_{0.1}^{0.4}$ | $1.0_{0.7}^{2.4}$ | $0.1_{0.1}^{0.1}$ | $0.9_{0.8}^{1.0}$ | $0.1_{0.1}^{0.2}$ | $1.0_{0.7}^{1.7}$ |
| $N_{7-13}^{11:00}$ [10$^3$cm$^{-3}$] | $19.7_{14.9}^{26.0}$ | $39.9_{15.0}^{53.9}$ | $8.1_{2.2}^{14.6}$ | $8.1_{4.6}^{12.7}$ | $3.9_{2.8}^{6.8}$ | $35.3_{10.6}^{63.3}$ | $0.2_{0.1}^{0.4}$ | $5.6_{3.0}^{10.8}$ |
| $N_{13-40}^{03:00}$ [10$^3$cm$^{-3}$] | $2.6_{1.6}^{3.2}$ | $4.5_{3.7}^{6.1}$ | $2.3_{1.2}^{3.8}$ | $4.8_{2.7}^{8.0}$ | $2.1_{1.4}^{3.3}$ | $2.8_{1.3}^{3.9}$ | $0.5_{0.3}^{1.3}$ | $3.3_{1.8}^{5.7}$ |
| $N_{13-40}^{12:00}$ [10$^3$cm$^{-3}$] | $29.0_{18.6}^{39.0}$ | $24.8_{18.9}^{40.3}$ | $9.3_{5.8}^{14.6}$ | $16.9_{10.0}^{22.5}$ | $10.6_{4.1}^{19.6}$ | $23.2_{9.3}^{34.5}$ | $0.7_{0.4}^{2.6}$ | $10.1_{6.3}^{13.3}$ |
| $N_{40-100}^{03:00}$ [10$^3$cm$^{-3}$] | $1.4_{1.0}^{2.7}$ | $3.3_{2.0}^{4.0}$ | $1.2_{1.0}^{1.9}$ | $2.8_{1.5}^{4.6}$ | $2.6_{2.2}^{3.8}$ | $1.8_{1.2}^{3.4}$ | $0.7_{0.4}^{0.9}$ | $1.3_{1.0}^{3.3}$ |
| $N_{40-100}^{17:00}$ [10$^3$cm$^{-3}$] | $6.0_{3.3}^{8.8}$ | $4.9_{2.8}^{6.5}$ | $2.2_{0.9}^{5.7}$ | $2.4_{1.7}^{3.4}$ | $3.3_{1.7}^{6.9}$ | $4.2_{2.9}^{5.8}$ | $0.6_{0.4}^{1.1}$ | $1.9_{1.2}^{3.0}$ |
| $eBC^{12:00}$ [$\mu g m^{-3}$] | $0.2_{0.1}^{0.5}$ | $0.2_{0.2}^{0.4}$ | $0.7_{0.3}^{1.0}$ | $0.9_{0.6}^{1.0}$ | $0.1_{0.1}^{0.2}$ | $0.3_{0.3}^{0.8}$ | $0.1_{0.0}^{0.4}$ | $0.5_{0.4}^{0.7}$ |
| $Sulph.^{med.\ day}$ [$\mu g m^{-3}$] | $0.6_{0.3}^{0.9}$ | $0.9_{0.5}^{1.3}$ | $0.3_{0.2}^{0.6}$ | $0.6_{0.3}^{0.8}$ | $1.9_{1.4}^{2.8}$ | $1.3_{1.2}^{1.7}$ | $0.2_{0.1}^{0.4}$ | $0.3_{0.1}^{1.2}$ |
| $CF^{14:00}$ [%] | $3_3^7$ | $3_3^3$ | $25_{10}^{91}$ | $15_4^{64}$ | $59_{38}^{81}$ | $12_6^{40}$ | $97_{45}^{97}$ | $84_{61}^{97}$ |
| $RH^{12:00}$ [%] | $31_{23}^{45}$ | $19_{13}^{27}$ | $48_{37}^{61}$ | $35_{30}^{40}$ | $61_{49}^{65}$ | $37_{34}^{41}$ | $75_{48}^{92}$ | $49_{42}^{53}$ |

Table 1. Key parameters at each category and station. Each entry shows the median value of the variable; first (Q1) and third (Q3) quartiles are denoted as subscripts and superscripts, respectively. For each variable subscripts and superscripts indicate the particle size range (where applicable) and the time of day or aggregation method used (e.g., 14:00 or max for maximum).