# Peer review of "New Particle Formation dynamics in the central Andes: Contrasting urban and mountain-top environments"

_Aerosol Research, 2024_

## Referee Comment (RC1)

Review for the manuscript: New Particle Formation dynamics in the central Andes: Contrasting urban and mountain-top environments by Diego Aliaga et al.

In this study measurements obtained during March 19 – May 31, 2018 (65 days), that correspond to the transition period from the wet to the dry season, are investigated for two sites in the Southern hemisphere (Bolivian Central Andes) including a mountain-top station (CHC) and an urban background site (EAC). The main objective of this work was to expand the current understanding of new particle formation (NPF) in the central Andes by simultaneously comparison of NPF events at the two interconnected sites (with different altitudes) and by additionally application of a recently developed "nanoparticle ranking analysis" (Aliaga et al., 2023) to evaluate the probability and intensity of NPF occurrence simultaneously at both sites, resulting in a joint distribution.

The manuscript is interesting, well written/ illustrated, describes careful experimental work and is worthy of being published in AR.

I have mostly the following minor revisions to suggest:

Minor comments:

In the abstract the four group categories are mentioned regarding the daily maximum concentration of 4-7 nm particles as high at both sites, lower at one etc. In the main text (lines 462-467) other parameters are also considered for naming the groups and this also should be stated. I would suggest adding those names (intense, cloudy etc.) in the abstract (lines 44-45) as well in parenthesis, as it is not very clear to the reader which is which.

For example: (A) high at both sites (intense NPF case), (B) medium at both (polluted), (C) high at EAC but low at CHC (volcanic), (D) and low at both (cloudy).

Also, in the abstract the D) cloudy related group is not mentioned at all. Maybe a line would be nice to be included.

Additionally, I would suggest for lines 461-468 to present the 4 cases as the classification, meaning, 1. intense, 2. polluted, 3. volcanic and 4. cloudy.

Lines 72, 321, 733 etc. A punctuation mark is missing.

Line 334. The parenthesis in reference is not necessary.

Line 547. The long — symbol should be changed to –.

Line 560. J4 should be $J_4$

Line 721. "EAC ( concentrations" should be "EAC (concentrations"

Line 725. The J4 values should be 2.2 and 9.2 cm-3 s-1for CHC and EAC, respectively.

Lines 881-882. Please rephrase. Not very clear to the reader.

Line 917. Add "days" after "Polluted"

In the introduction authors state:

"For example, we need to better understand the chemical composition of the precursor gases involved in nucleation and growth, as well as the composition of the aerosols formed during these events." But this is not addressed in the current work.

References.

Aliaga, D., Tuovinen, S., Zhang, T., Lampilahti, J., Li, X., Ahonen, L., Kokkonen, T., Nieminen, T., Hakala, S., Paasonen, P., Bianchi, F., Worsnop, D., Kerminen, V.-M., and Kulmala, M.: Nanoparticle ranking analysis: determining new particle formation (NPF) event occurrence and intensity based on the concentration spectrum of formed particles, Aerosol Res., 1, 81‑92, https://doi.org/10.5194/ar-1-81-2023, 2023.

---

## Author Comment (AC1)

In the following, reviewer comments are shown in **green**, and our replies to these comments are shown in **black** and modifications to the original text in the manuscript are shown in **red**.

**RC1**

Review for the manuscript: New Particle Formation dynamics in the central Andes: Contrasting urban and mountain-top environments by Diego Aliaga et al.

In this study measurements obtained during March 19 – May 31, 2018 (65 days), that correspond to the transition period from the wet to the dry season, are investigated for two sites in the Southern hemisphere (Bolivian Central Andes) including a mountain-top station (CHC) and an urban background site (EAC). The main objective of this work was to expand the current understanding of new particle formation (NPF) in the central Andes by simultaneously comparison of NPF events at the two interconnected sites (with different altitudes) and by additionally application of a recently developed "nanoparticle ranking analysis" (Aliaga et al., 2023) to evaluate the probability and intensity of NPF occurrence simultaneously at both sites, resulting in a joint distribution.

The manuscript is interesting, well written/ illustrated, describes careful experimental work and is worthy of being published in AR.

We thank the reviewer for the positive feedback and constructive comments which we address below.

I have mostly the following minor revisions to suggest:

**Minor comments:**

1. In the abstract the four group categories are mentioned regarding the daily maximum concentration of 4-7 nm particles as high at both sites, lower at one etc. In the main text (lines 462-467) other parameters are also considered for naming the groups and this also should be stated. I would suggest adding those names (intense, cloudy etc.) in the abstract (lines 44-45) as well in parenthesis, as it is not very clear to the reader which is which.

For example: (A) high at both sites (intense NPF case), (B) medium at both (polluted), (C) high at EAC but low at CHC (volcanic), (D) and low at both (cloudy).

Also, in the abstract the D) cloudy related group is not mentioned at all. Maybe a line would be nice to be included.

Thank you for the suggestion, we have changed the abstract to follow the suggested format. We now include the category names and a sentence about category D (now referred as 4). The abstract now reads as follows:

In this study, we investigate atmospheric new particle formation (NPF) across 65 days in the Bolivian Central Andes at two locations: the mountain-top Chacaltaya station (CHC, 5.2 km above sea level) and an urban site in El Alto-La Paz (EAC), 19 km apart and at 1.1 km lower altitude. We  classified the days into four categories based on the intensity of NPF, determined by the daily maximum concentration of 4-7 nm particles: (1) high at both sites, (2) medium at both, (3) high at EAC but low at CHC,  and (4) low at both. These categories were then named after their emergent and most prominent characteristics: (1) Intense-NPF, (2) Polluted, (3) Volcanic, and (4) Cloudy. This classification was premised on the assumption that similar NPF intensities imply similar atmospheric processes. Our findings show significant differences across the categories in terms of particle size and volume, sulfuric acid concentration, aerosol compositions, pollution levels, meteorological conditions, and air mass origins. Specifically, intense NPF events (1) increased Aitken-mode particle concentrations (14-100 nm) significantly on 28% of the days when air masses passed over the Altiplano. At CHC, larger Aitken-mode particle concentrations (40-100 nm) increased from $1.1 \times 10^3$ cm$^{-3}$ (background) to $6.2 \times 10^3$ cm$^{-3}$ very likely linked to the ongoing NPF process. High pollution levels from urban emissions on 24% of the days (2) were found to interrupt particle growth at CHC and diminish nucleation at EAC. Meanwhile, on 14% of the days, high concentrations of sulphate and large particle volumes (3) were observed, correlating with significant influences from air masses originating from the actively degassing Sabancaya Volcano and a depletion of

positive 2-4 nm ions at CHC.  but not at EAC. During these days, reduced NPF intensity was observed at CHC but not at EAC. Lastly, on 34% of the days, overcast conditions (4) were associated with low formation rates and air masses originating from the lowlands east of the stations. In all cases, event initiation (~9:00) generally occurred about half an hour earlier at CHC than at EAC and was likely modulated by the daily solar cycle. CHC at dawn is in an airmass representative of the regional residual layer with minimal local surface influence due to the barren landscape. As the day progresses, upslope winds bring in air masses affected by surface emissions from lower altitudes, which may include anthropogenic or biogenic sources. This influence likely develops gradually, eventually creating the right conditions for an NPF event to start. At EAC, the start of NPF was linked to the rapid growth of the boundary layer, which favoured the entrainment of air masses from above. The study highlights the role of NPF in modifying atmospheric particles and underscores the varying impacts of urban versus mountain-top environments on particle formation processes in the Andean region.

2. Additionally, I would suggest for lines 461-468 to present the 4 cases as the classification, meaning, 1. intense, 2. polluted, 3. volcanic and 4. cloudy.

Thank you for the suggestion. We have now reordered the bullet points in the suggested order and numbered in agreement with the abstract.

3. Lines 72, 321, 733 etc. A punctuation mark is missing.

Thank you. This has been corrected now.

4. Line 334. The parenthesis in reference is not necessary.

Thank you. This is corrected now.

5. Line 547. The long — symbol should be changed to –.

Thank you. This is corrected now.

6. Line 560. J4 should be $J_4$

Thank you. This is corrected now.

7. Line 721. "EAC ( concentrations" should be "EAC (concentrations"

Thank you. This is corrected now.

8. Line 725. The J4 values should be 2.2 and 9.2 cm-3 s-1for CHC and EAC, respectively.

Thank you. This is corrected now.

9. Lines 881-882. Please rephrase. Not very clear to the reader.

Thank you for the suggestion. The text has now been rephrased and reads as follows:

We observed that intense NPF events (Intense-NPF)  occur under different atmospheric conditions at each site. At EAC, they begin in a rapidly growing PBL , where the entrainment of cleaner air from above mixes with polluted air from local emissions. At CHC, they start in a regional residual layer , increasingly influenced by regionally surface -influenced airmasses due to anabatic winds. In both cases, air masses  from the Altiplano.  play a key role, with the events  eventually merging to form a regional mixture across the PBL. This indicates that the formation of NPF events is not confined to specific atmospheric layers but rather involves complex interactions across different layers of the atmosphere. We do not observe any event in the free troposphere as during event days and in most of the time except for brief periods at dawn, our air mass analysis suggests that CHC is heavily influence by the residual layer.

10. Line 917. Add "days" after "Polluted"

Thank you. This is corrected now.

11. In the introduction authors state:

"For example, we need to better understand the chemical composition of the precursor gases involved in nucleation and growth, as well as the composition of the aerosols formed during these events." But this is not addressed in the current work.

We have added text to the end of the introduction to specify that in this study we only address the role of sulfuric acid. The mentioned text now reads:

In this study we expand our current understanding of NPF in the central Andes through the analysis of 65 days of observations (March 19 – May 31, 2018) at CHC and EAC that correspond to the transition period from the wet to the dry season. We apply the recently developed "nanoparticle ranking analysis" (Aliaga et al., 2023) to evaluate the probability and intensity of NPF occurrence simultaneously at both sites, resulting in a joint distribution. This analysis is integrated with an in-depth examination of air mass histories at CHC (Aliaga et al., 2021) and EAC, and complemented with observations of aerosol chemical composition concentrations (Bianchi et al., 2022). Although precursor gases were not measured at EAC, we include direct sulfuric acid measurements at CHC, where sulfuric acid–ammonia interactions are likely the primary mechanism for NPF (Zha et al., 2023a). Our study aims to deepen the understanding of NPF in the Andes, focusing on both background and polluted regimes. We investigate the regional representativeness (both vertical and horizontal) of NPF observations at CHC and their correlation with events at EAC. We also quantify the impact of anthropogenic emissions at EAC, locally and at a sub-regional level. Moreover, we identify local and regional meteorological patterns and air transport mechanisms that influence NPF, assessing their effects on both sites. Lastly, we evaluate the contrasting influence of volcanic plumes on NPF at CHC and EAC.

**RC2**

Review for the manuscript: New Particle Formation dynamics in the central Andes: Contrasting urban and mountain-top environments by Diego Aliaga et al.

This study investigates the Atmospheric New Particle Formation (NPF) in the Central Andes, focusing on differences between mountain-top and urban sites. This study examines measurements taken at two sites in the Southern Hemisphere (Bolivian Central Andes): a mountain-top station (CHC) and an urban background site (EAC). The measurements were taken from March 19 to May 31, 2018, which corresponds to the dry season's transition. Classification of study days into four categories based on NPF intensity for detailed analysis. Identification of key indicators and detailed analysis of categories to understand NPF processes in the region, especially the applicability of "nanoparticle ranking analysis" proposed by Kulmala et al. (2022). It provides valuable insights into the factors influencing particle formation, particularly in exploring the impacts of urban and mountain environments on NPF processes. The classification of study days into distinct categories based on NPF intensity allows for a comprehensive analysis of atmospheric conditions and their impact on NPF, shedding light on the complex interplay between various factors. Nevertheless, before it is published, the manuscript needs considerable refinement.

We thank the reviewer for the positive feedback and constructive comments which we address below.

**Comments:**

1. The present abstract ought to delve into the dynamics of the mountain-valley wind in a diurnal cycle, given its crucial role in the NPF process at elevated altitudes.

Thank you for the suggestion. We have added the following text to the abstract:

In all cases, event initiation (~9:00) generally occurred about half an hour earlier at CHC than at EAC and was likely modulated by the daily solar cycle. CHC at dawn is in an airmass representative of the regional residual layer with minimal local surface influence due to the barren landscape. As the day progresses, upslope winds bring in air masses affected by surface emissions from lower altitudes, which may include anthropogenic or biogenic sources. This influence likely develops gradually, eventually creating the right conditions for an NPF event to start. At EAC, the start of NPF was linked to the rapid growth of the boundary layer, which favoured the entrainment of air masses from above.

2. Some insight into the significance of the findings from the mountain observations should be provided in the introduction section. in particular over India's Western Ghats (Victor et al., 2024, Sebastian et al., 2022)

Victor N J, Pallavi Buchunde, Mathew Sebastian,Vijay P. Kanawade, Devendraa Siingh, Subrata Mukherjee, Swapnil S. Potdar, T. Dharmaraj, and Govindan Pandithurai (2024), Characteristics of new particle formation events in a mountain semi-rural location in India, Atmospheric Environment, Volume 324, 1 May 2024, 120414, https://doi.org/10.1016/j.atmosenv.2024.120414

Sebastian, M., Kanawade, V.P., Soni, V.K., Asmi, E., Westervelt, D.M., Vakkari, V., Hyv¨arinen, A.P., Pierce, J.R., Hooda, R.K., 2021a. New particle formation and growth to climate-relevant aerosols at a background remote site in the western himalaya. J. Geophys. Res. Atmos. 126 https://doi.org/10.1029/2020JD033267.

We thank the reviewer for the suggestion. We have now included the suggested references along with other relevant studies at mountaintop stations and or two-site (or multisite) studies. The modified text reads as follows:

Despite the growing number of global observations, there is a pronounced bias toward studies conducted in the Northern Hemisphere and at lower altitudes, leaving the Southern Hemisphere and high-altitude (urban or background) regions underrepresented in NPF research (Laj et al., 2020; Nieminen et al., 2018). High-altitude studies in locations such as the Himalayas, the Swiss Alps, and the Rocky Mountains have provided important insights into NPF processes under these unique atmospheric conditions (e.g., Bianchi et al., 2016, 2021; Cai et al., 2021; Hirshorn et al., 2022; Lv et al., 2018; Nishita et al., 2008; Sebastian et al., 2021; Tang et al., 2023; Victor et al., 2024; Sellegri et al., 2019 and references therein; Kerminen et al., 2018 and references therein), but these are still relatively limited in number.

…

Two-site studies combining mountaintop and ground-level measurements have been beneficial in revealing how elevation, local topography and emissions influence NPF dynamics and the transport of air masses (e.g., Boulon et al., 2011; Casquero-Vera et al., 2020; Shang et al., 2023; Wang et al., 2014; Zhou et al., 2021).

3. Line 70: Kerminen et al., 2018 and Lee et al., 2019 may be cited

Lee, S.-H., Gordon, H., Yu, H., Lehtipalo, K., Haley, R., Li, Y., & Zhang, R. (2019). New particle formation in the atmosphere: From molecular clusters to global climate. Journal of Geophysical Research: Atmospheres, 124, 7098–7146. https://doi.org/10.1029/2018JD029356

Thank you. The references have been added now.

4. 110: There have been reports that indicate mixed air leads to more NPF, while clear air also occasionally delivers greater NPF. How does this fit into this paragraph?

Thank you for the observation. We have now added text related to the NPF heterogeneity within the convective boundary layer to address the raised issue. The paragraph now reads as follows:

Due to the nature of NPF events, which last for several hours and are strongly influenced by the path and history of the involved air masses involved, any NPF-related observation at a fixed point in space will depend strongly depends on the origin and path thatwhere the air mass has taken through the atmospheretraveled before arriving atreaching the measurement pointsite. Whether the air mass has passed over polluted urban areas with high emissions or has or descended from the clean upper troposphere will influence the affect its chemical and physical properties of that. Furthermore, the NPF process in the convective boundary layer is heterogeneous, influenced by variations in thermodynamic conditions, precursor gas availability, pre-existing aerosol populations, and the continuous yet not instantaneous air mass mixing within the vertical column, which typically takes around 30 minutes (O'Donnell et al., 2023). For example, at the upper limit of the boundary layer, lower temperatures can reduce the volatility of nucleating vapors, enhancing nucleation, while intrusions from the free troposphere may decrease the condensation sink, increasing the survival probability of newly formed clusters. Near the surface, emissions of precursor gases may favor nucleation, though pre-existing aerosol and primary particle emissions can also raise the condensation sink, reducing the survival probability of clusters (O'Donnell et al., 2023)this reason,. Despite this heterogeneity, continuous mixing over time can make the process appear homogeneous when observed hours after the event has begun. These considerations

highlight the importance of accurately tracing the geographical history of the air mass before  sampling . Additionally, multisite observations can help disentangling the various factors influencing heterogeneous NPF.

5. Line 215-220: NAIS measurement methods can be made shorter by including appropriate citations, as this information is widely available.

Thank you for the recommendation. We have now reduced the text and merged the paragraph with the following in order to streamline the text. It now reads as follows:

The NAIS (Mirme and Mirme, 2013) nominally measures the number size distributions of ions in the electric mobility diameter range between 0.~~842 nm and 42 nm and particles in the range between 2.5 and 42 nm. The NAIS operates with two parallel measurement columns, each dedicated to a specific polarity. During ion measurements, positive and negative ions are simultaneously measured in their respective columns. For particle measurements, aerosol particles are charged to opposite polarities using corona chargers and then measured concurrently in their respective columns. It is important to note that particle observations below about 2.5 nm are contaminated by charger ions and therefore are excluded from the measured size range for particles. The separation of air ions and charged particles is based on their electrical mobilities, with detection carried out by a multichannel differential mobility analyzer (DMA).~~84 nm and 42 nm and particles in the range between 2.5 and 42 nm.

6. If details have already been mentioned in Aliaga et al., (2021), then avoid repetition or compress if necessary for the reader. Simple statements, elaborating less significant details, and repetition of the same citation may be avoided by taking the journal's standard and the length of the MS into consideration. For example, lines 350, 335 may be avoided.

Thank you for the suggestion. We have streamlined the section trying to avoid repetition and focusing on mentioning only what is different in this study compared to Aliaga et al. (2021). The relevant section now reads as follows:

All  air mass history analyses in this study are derived from 4-day backward simulations using the Lagrangian FLEXible PARTicle dispersion model (FLEXPART; version FLEXPART-WRF_v3.3.2; Brioude et al., 2013), with CHC and EAC as the arrival points . The FLEXPART simulations were driven by meteorological output from the Weather Research and Forecasting (WRF) model version 4.0.3 (Skamarock et al., 2019). For CHC, we utilized a pre-existing dataset  generated by Aliaga et al. (2021), while for EAC, we ran new FLEXPART simulations using an identical setup, but targeting EAC as the arrival point.

The WRF meteorological data were originally generated by Aliaga et al. (2021) for the CHC region during the SALTENA campaign (2017-12-06 to 2018-05-31) at high resolution (down to 1 km). The WRF model was initialized with data from the National Centers for Environmental Prediction Climate Forecast System Version 2 (Saha et al., 2011, 2014) and included four nested domains, with the innermost covering 180 km$^2$ at 1 km resolution. The model output was saved every 15 minutes, and further details on the simulation setup are available in Aliaga et al. (2021).

In both the CHC and EAC simulations, 20,000 virtual particles (passive air tracers) were released every hour and tracked backward for four days. To refine the results from Aliaga et al. (2021), which used a pseudo PBL altitudes below 1.5 km, we employed a more precise PBL metric. We determined whether air masses were within or above the PBL by comparing their geographic positions and altitudes against the PBL height in the WRF meteorological dataset.

~~, produced a high-resolution (up to 1 km) meteorological data set for a large area around CHC and for the 6-month duration (2017-12-06 to 2018-05-31) of the SALTENA campaign by running the Weather Research and Forecasting (WRF) model version 4.0.3 , which is a state-of-the-art, non-hydrostatic, regional numerical weather prediction model. The initial and boundary conditions were taken from the National Centers for Environmental Prediction Climate Forecast System Version 2 . Four nested domains were included, with the outermost domain covering large parts of South America and the Pacific Ocean with a grid spacing of 38 km and the innermost domain covering an~~

area of approximately 180 km by 180 km centered on CHC with a grid spacing of 1 km. The model output was saved every 15 minutes. Further details of the WRF model simulation set up are given in .

also ran FLEXPART simulations to identify the origins of air masses arriving at CHC. For the campaign's duration, 20,000 virtual particles (virtual passive air tracers) were released every hour and tracked backwards for four days before their arrival at CHC. In this study, we utilize this pre-existing dataset and additionally, we also performed new FLEXPART simulations to identify the origins of air masses arriving at EAC. These additional FLEXPART simulations were also driven by the WRF model output from  and again 20,000 virtual particles were released every hour.

When run in backwards mode, FLEXPART computes the emission sensitivity response function, also referred to as the source–receptor relationship (SRR), on a user-specified three-dimensional longitude–latitude–height grid (Eckhardt et al., 2017; Pisso et al., 2019; Seibert and Frank, 2004). The potential emission sensitivity provides a footprint of emission source areas and if multiplied by actual emissions would give an estimate of the concentrations that would be measured at the receptor (station). The SRR output was casted into a log polar grid centered at each of the stations following the methods and rationale described by Aliaga et al. (2021)

7. Line no. 390-395 may be removed, the mode-fitting method is quite known, and sufficient citation also provided. Moreover, as mentioned, it is not suitable for the present study.

Thank you for the recommendation. We have removed the text regarding the mode-fitting method. The relevant paragraph now reads as follows:

The growth rate (GR) indicates the time-based change in diameter (D) of the growing mode in the PNSD. GR can be estimated in field studies using the mode-fitting, maximum-concentration or, appearance time or mode-fitting methods (Kulmala et al., 2012; Lehtipalo et al., 2014). The mode-fitting method fits a log-normal mode to the particle size distribution at each time step of a formation event and uses the geometric mean diameter for the growing mode, with GR calculated from the slope of a least-squares line through these means. The maximum concentration method identifies peak concentrations at each size bin after applying a Gaussian filter for noise reduction and fits a least-squares line to these peaks with swapped axes (x-axis is diameter, y-axis is time) since the uncertainty is in the time dimension, calculating GR as the inverse of this slope. The appearance time method is similar to the maximum concentration method but instead of using the maximum, a certain threshold of the maxima (e.g. 50%) is used. In this study we use the maximum concentration method (Max) and the appearance time method setting the threshold as the maximum of the time derivative (Der.). We do not use the mode-fitting method because events in this region nucleate for many hours making the method unsuitable for this environment.

8. Given the number of analyses, parameters, and presumptions used here, it is strongly advised to employ a flow chart with a few brief descriptions. This will help the reader to better grasp the technique and keep the text to a minimum.

Thank you for the suggestion. We are assuming that the reviewer refers to the method for grouping the days into categories. In this regard, we have added the following flowchart to the manuscript:

[Figure]

Figure 2: Flowchart showing the procedure for assigning each day to one of the four categories that are subsequently evaluated and characterized.

9. Line number 475 to 490 also be represented as a chart or table, so that the reader can easily understand its significance.

Thank you for the suggestion. We are now representing the suggested text in the following table that has been added to the manuscript:

| Size range [nm] | Notation | Description |
|---|---|---|
| 2–4 | $N_{2-4}^{-\,(+)}$ | • Primarily observes reliable negative (positive) ion concentrations
• Indicates early clustering and growth
• Marks the onset of nucleation |
| 4–7 | $N_{4-7}$ | • Falls within the lower nucleation range
• Sensitive to the intensity of new particle formation (NPF)
• Marks early nucleation and growth stages |
| 7–13 | $N_{7-13}$ | • Represents the upper nucleation range
• Reflects further growth
• Not influenced by primary emissions
    at EAC during morning and afternoon rush hours |
| 13–40 | $N_{13-40}$ | • Corresponds to the lower Aitken mode
• Most NPF–produced particles are
    found in this range by the end of daily NPF events
• Captures traffic primary emissions at EAC |
| 40–100 | $N_{40-100}$ | • Pertains to the upper Aitken mode
• A small but significant portion of NPF–born particles reach this size
• Sensitive to traffic primary emissions at EAC |
| 100–440 | $N_{100-440}$ | • Denotes the accumulation mode
• Affected by primary emissions from pollution, dust
• Notably influenced by volcanic
    plume influences in this specific geographic area |

Table 2: Particle size ranges used in this study and their characteristics.

10. It seems the cloud cover or fraction between CHC and EAC varied much, for example, SW at CHC shows the drift from the normal curve is more visible much later, of 12 LT and the temperature at EAC doesn't seem to differ much from the average curve. Hence, in the in the morning hours, where nucleation is likely to happen, at both sites, the atmosphere was exposed to either a clear sky or enough sunlight or solar radiation for the photochemical activity. Moreover, CS and J4 at EAC for intense-NPF and cloud cover-NPF almost have same pattern, so the question comes what the impact of cloud cover at EAC is. So to clear this uncertainty, it would be great if hourly cloud fraction data to be presented.

Thank you for your suggestion and observation. We assume the reviewer meant that the cloud cover fraction between CHC and EAC did not vary much. Under this assumption, we agree that the drift in SW at CHC from the median curve becomes more pronounced later than 12 LT in the Cloudy category, and that temperature at EAC shows little deviation from the median curve.

In the manuscript, we have presented satellite cloud cover fraction data from MODIS at 02:00, 10:00, 14:00 and 22:00 LT (the only four daily overpasses). These are shown in Table 2 (previously only at 14:00 LT, but now also at 10:00 LT), and in Figures S6 and S9. At EAC, the 10:00 cloud fraction values differ significantly between the Intense-NPF (3%) and Cloudy (29%) categories. Similarly, at CHC, the values are 3% for Intense-NPF and 62% for the Cloudy category. Therefore, while the atmosphere was exposed to solar radiation in the Cloudy category, it was significantly less compared to the Intense-NPF category.

Regarding the comparison of J4 and CS at EAC between the Intense-NPF and Cloudy categories, there may be some misunderstanding. J4 at EAC shows a significant difference: 50.7 cm$^{-3}$s$^{-1}$ for Intense-NPF versus 6.1 cm$^{-3}$s$^{-1}$ for the Cloudy category. Therefore, we disagree that they have the same pattern. This also suggests cloud cover may have a substantial impact on formation rates, though cloud cover is not the only factor—differences in water vapor mixing ratio and air mass origin also play a role.

In summary, while we cannot provide hourly cloud fraction data, we do present cloud fraction values at four times during the day. To emphasize the morning values (10:00), we have now included them in Table 2.

11. In conclusion: In addition to the fact that all of the measures and numbers presented have already been covered in previous sections, they may be constrained to focus solely on the conclusions drawn from the metrics. This makes it possible for readers to focus solely on the study's primary findings.

Thank you for the suggestion. We have decided to remove these values that have been already presented. The relevant text now reads as follows:

We obtained four significantly different categories of days :

named based on their emergent characteristics at both sites: (1) Intense-NPF, (2) Polluted, (3) Volcanic, and (4) Cloudy.

12. In conclusion, a small paragraph may be devoted to summarizing the findings of the entire investigation in relation to the high-altitude readings.

Thank you for the recommendation. We have included now a concluding paragraph and it reads as follows:

To conclude, NPF in this region is frequent and intense compared to other locations around the globe (Kerminen et al., 2018; Sellegri et al., 2019), occurring in both urban polluted environments and at mountaintop sites. It is primarily driven by air masses originating from or passing over the Altiplano, coinciding with synoptic conditions that favour clear skies and dry conditions. In contrast, NPF is significantly diminished when air masses originate from the lowlands (Amazon and Chaco). At the mountaintop, the main driving mechanism is likely sulfuric acid-ammonia nucleation (Zha et al., 2023a), while in the city, local emissions enhance formation rates possibly through amines or other basic stabilizers. At both sites, events are typically observed uninterruptedly throughout the day and consistently reach sizes between 40 and 100 nm.